# The Relationship between Urban Diversity and Residential Segregation

Robert William Pendergrass 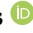

TallgrassGIS, Broken Arrow, OK 74012, USA; rwpendergrass@tallgrassgis.com; Tel.: +1-918-706-1523

**Abstract:** Racial diversity was found to be related to racial residential segregation and strongly related to racial isolation within the nation's metropolitan and micropolitan areas at the block group level. However, the relationships were both complex and dependent on the racial group. Racial diversity was assessed for all 927 metropolitan and micropolitan areas as opposed to just the largest fifty or the largest one hundred. Racial segregation and isolation were assessed at the block group level (excluding water and zero population block groups), not the census tract level, within each metro/micro area. The eight non-overlapping racial groups as defined by the U.S. Census were used. Racial diversity was measured with the Diversity Index (the Simpson Index). Racial residential segregation was measured with the pairwise Dissimilarity Index (D) and the Multigroup Dissimilarity Index ($D_G$) as it was initially proposed using expected frequencies. Racial isolation was measured with the Isolation Index (P*).

**Keywords:** segregation; diversity; race; demographics; isolation; multigroup segregation index ($D_G$); pairwise segregation index (D); isolation index (P*); diversity index (DI)





## 1. Introduction

The purpose of this research was to determine the relationship between unban racial diversity and urban racial residential segregation in the United States. Research on racial segregation has long been a focus in the social sciences. While often mentioned in an offhand manner, the topic of diversity has become increasingly prevalent in recent decades with workplace and other equity concerns [1–4].

The balance of this introduction establishes definitions of the concepts employed in the research: urban areas; differentiation—diversity and differentiation—segregation. Section 2, Data and Methods, presents the racial data, the geographic units of analysis, operation definitions of diversity and operational definitions of residential segregation. The section presents the research hypothesis and three operational hypotheses with corresponding null hypotheses.

Section 3 presents the results. First for the overall distribution of the diversity indices is discussed along with the geographic distribution. Secondly, each of the segregation indices are discussed and their geographic distribution presented: multigroup dissimilarity; pairwise dissimilarity and isolation. For the four major racial categories (black, Hispanic, Native American, and Asian) the overall distribution of the dissimilarity and isolation indices are presented as well as their geographic distribution. lastly presented is the empirical analysis of the relationship between racial diversity indices and the multigroup dissimilarity indices and the relationship between racial diversity indices and each of the pairwise dissimilarity indices.

Section 4 presents the conclusions of the research.

### 1.1. Urbanism-Metropolitanism

Racial segregation, typically discussed in an urban context, is equally as significant in "rural" communities as well as the larger metropolitan areas [5,6]. Research, discussion, and comprehension of racial segregation therefore requires an accurate definition of "urban".

Human settlements have been classified by size, economic function, and physical footprint. Settlements are on a continuum, starting with the isolated residence and increasing in size and functions from a crossroad, a hamlet, a small village, a large village, a town, a city, to a megalopolis. Nevertheless, the question of what is urban has plagued social research due to its consequences on the structure of a study about the units of analysis. Researchers have been forced to focus on areas with available data. There are two aspects of "urban" or the "city": the actual physical scope of the city and its labor market. The physicality of the urban area is the core underlying view of the city, and researchers try to approximate the physical city as best as possible. To be urban, usually, three criteria must be met: population size, density, and physical compactness or contiguousness.

Much research on segregation has been on the more significant incorporated places: the incorporated City of Chicago, the incorporated City of New York, and the incorporated City of Washington DC. In the past, the legally incorporated area of a city did match the physical city reasonably well. However, continued population growth has outstripped the political boundaries of most cities.

The best approximation of settlement patterns on the ground is the Urbanized Area defined by the U.S. Bureau of the Census. It is the best systematic attempt to empirically identify a physical city. Urbanized Areas ignore all political boundaries. An urbanized area is the core of all Standard metropolitan Area.

The Metropolitan Statistical Area (MSA) is a much better approximation of the economic "city" and its labor market, but it includes a great deal of undeveloped or agricultural land. The MSA was not designed to account for the physical aspect of a city, being more of a representation of the labor market and commuting area of the core urban area: its economic footprint. (Ratcliffe n.d.) With the advent of the Metropolitan Area concept, the reliance has switched to the Metropolitan Statistical Area of Chicago, the Metropolitan Statistical Area of New York, and the Metropolitan Statistical Area of Washington DC. See the discussion by Michael Ratcliffe [7] on the 100 years of trying to define "Urban" by the U.S. Bureau of the Census.

### 1.2. Differentiation—Diversity

Both diversity and segregation are distinct aspects of the same phenomenon—differentiation. Diversity is the non-spatial aspect, while segregation is the spatial aspect.

The concepts of homogeneity and heterogeneity are amazingly nebulous terms, though intuitively, individuals know what they mean. Lineberry and Fowler in 1969 wrote, "homogeneity [and heterogeneity] is easily one of the most ambiguous terms in the ambiguous language of the social sciences". Patil and Tallie, in a discussion of ecology, add that diversity itself has weak conceptual foundations [8]. However, its importance is stated by Louis Worth, who indicates that diversity is one of the defining characteristics of a city: "For sociological purposes a city may be defined as a relatively large, dense, and permanent settlement of socially heterogeneous individuals" [9]. Sugihara adds that the "importance of [. . . ] diversity hinges directly on its possible connection with the functioning and organization of communities" [10]. Sugihara also adds ". . . how little is known about the connection between diversity patterns and community processes, . . . " [10].

Diversity embodies the notion of variety. Any population can be divided up into many other groups. Diversity refers to the variety that results from the differentiation of a characteristic that can divide a population into distinct groups on a given dimension or attribute. Examples include age, occupation, gender, race, income level, ethnicity (as in the country of origin), and educational attainment. Diversity is aspatial or non-spatial.

Diversity is on a continuum running between homogeneity and heterogeneity, with the terms applying to the opposite ends (Figure 1).

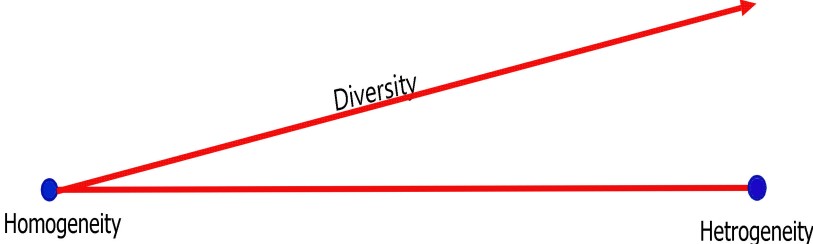

**Figure 1.** The Diversity Continuum.

Joel Smith, in looking at the issue of creating homogeneous census tracts and neighborhoods, defined homogeneity as "... a property of an area such that the distribution of specified population characteristics within that area will be found to exist (with minor deviation) in the population contained in any segment chosen from within it" [11]. This definition is confined to a physical area such as a city or census tract. The condition can also apply to the population itself though it has an obvious border constraint.

These definitions apply:

a.  Homogeneity can be defined as the condition or state to which the members of a population belong to, and are in, the same category or background factor. It is the condition where diversity is zero.

b.  Heterogeneity can be defined as the condition or state in which members of a population belong to, or in, distinct categories of the same background factor, that is, the absence of homogeneity. It is the condition where diversity is non-zero, the more diversity, the greater the heterogeneity.

Typical usage of the term diversity is synonymous with heterogeneity: the greater the heterogeneity, the greater the diversity. For the eight categories of race used here, if 100 percent of the population were within one category, for example, all are non-Hispanic white alone, and the other seven categories held no one, there would be a state of complete homogeneity. If the population was evenly split across all eight categories, there would be a state of total heterogeneity.

There are multiple dimensions or background factors required for a community to be homogeneous on numerous dimensions simultaneously (at least theoretically): racially, income, occupation, religion, and so on. A tight-knit community like the Amish would come close to this situation. The author adds these definitions:

a.  Community Homogeneity can be defined as a condition, or state, in which the members of a population belong to and are in the same category within each background factor, that is, in a state of homogeneity for each identified dimension or background factor.

b.  Community heterogeneity can be defined as a condition or state in which the members of a population belong to and are in distinct categories within each background factor, that is, the absence of community homogeneity.

*1.3. Differentiation—Residential Segregation*

Most studies have dealt with the mechanics of measuring segregation while lacking conceptual foundations of segregation.

Similar to diversity, spatial differentiation is on a continuum but refers to the groups' spatial dispersion (Figure 2). The difference is on which end of the continuum is referenced. Here, it is on the segregated side where all the members of one group are in the same space. Segregation itself refers to the spatial separation of the population on a given trait or characteristic from others, including the dominant group. It is maintained by social, political, and economic processes, and it is usually an involuntary separation. Morrill defines segregation as "the spatial separation of groups, resulting from certain physical or social processes" [12]. Massey, Rothwell, and Domina define it as "... the separation of socially defined groups in space, such that members of one group are disproportionately concentrated in a particular

set of geographic units compared with other groups in the population" [13]. White adds that in a sociological sense, "… segregation means the absence of interaction between groups". In a geographic sense, it means the "… unevenness in the distribution of social groups in physical space" [14].

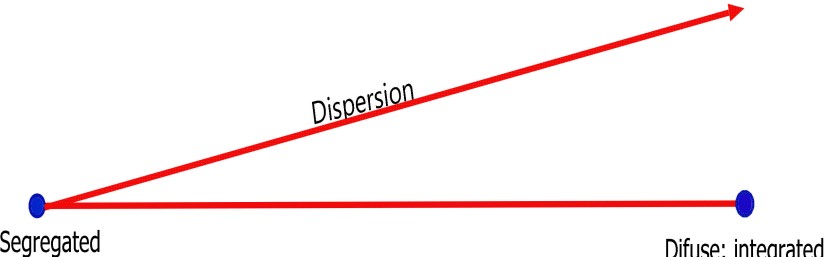

**Figure 2.** The Segregation Continuum.

Segregation becomes important because "individuals of various groups occupy and experience different social environments" [15]. Morrill adds that "Segregation is not accidental, but a consequence of purposeful behavior. The motivations for segregation are structural, that is, a desire to minimize interaction with certain other kinds of people.

The most effective tool or manifestation is territorial separation" [12]. Massey adds, "… segregation makes racial subordination efficient because it confines blacks to identifiable areas so that disinvestment in a place brings disinvestment in a people" [15].

The greater the dispersion of a group across space, the more integrated or diffuse it is. The less dispersion of a group, the more segregated and isolated it is.

Segregation is a multi-faceted concept. The idea of physical space is inherent in the idea of segregation. Segregation is the physical separation of the population on a given characteristic; how mixed are the groups spatially. Spatial differentiation becomes the question. How differentiated are neighborhoods based on race? Ethnicity? Religion? Socio-economic status?

Residential racial segregation has not been and is not a random process, and once in place, it becomes entrenched and is difficult to reverse. There is a level of residential segregation that is voluntary. Residents choose to live in an area that contains other residents who are like them. This voluntary process results in enclaves and clusters. However, this baseline is indeterminant. There have been some suggestions that the baseline can be estimated to be what the distribution would be if randomized. Choosing where one is to live is not a random process.

Residential racial segregation has historically been established and maintained by mechanisms and social processes which make the choice of where to live involuntary. Even if one thinks his or her decision as to where to live is voluntary, it really is due to subconscious influences. Many residents are forced to live in a particular area (or areas) directly or indirectly. Most discussions and research deal with involuntary segregation. Once established, how/why segregation is maintained becomes the real question.

In Cycles of Segregation [16], the process of residential choice was examined. They present several factors that have been offered as explanations in addition to methods used to cause segregation. They are used to enforce segregation which was also discussed by Elizabeth Bruch [16] and by Camille Charles [17]:

1. Differences in household residential preferences include the desire to live among one's own racial group (basically, this is voluntary segregation). The extreme position is the hypothesis that the within-group (blacks, Hispanics, or whites) preferences explain all segregation.

2. Individual and group discrimination against the out-group. The same-race preference hypothesis leaves out the converse that the dominant in-group keeps the minority group out (i.e., discrimination). This includes, among others, racial stereotyping and avoiding neighborhoods due to their racial composition. Charles states:

*"The overall conclusion to be drawn is that active racial prejudice is a critical component of preferences for integration, and therefore, the persistence of racially segregated communities. Whites' racial prejudice is a double whammy: influential not only for its effect on their own integration attitudes but also for its implications for minority group preferences and residential search behavior".* [17]

Charles's work reinforces the work by Eleanor Wolf of the impact of discrimination and prejudice on residential segregation in the discussion about a preference threshold, a leaving threshold and a willingness-to-enter threshold for both whites and blacks. The predominant issue were the attitudes whites [18].

3. Institutional discrimination in many forms. Local biased zoning provisions, discriminatory protective covenants, discriminatory lending practices both by private lenders and in federal housing programs; discriminating real estate practices (steering blacks to predominately black neighborhoods and the converse for whites, blockbusting, etc.). See [19] for maps of redlining in large cities. Often forgotten roadblocks have been restrictive racial covenants [20]. These serve to block the minority group from entering a neighborhood. This type of situation has been termed the place stratification hypothesis.

4. Differences in access to socio-economic resources (termed the spatial assimilation hypothesis). Research has established that educational status and income status have a small role and little to do with racial segregation [16]. Charles points out that blacks do not see the same payoff for improved social status as do Asians and Hispanics [17] and adds, "The oppositional experiences of blacks and whites contradict the tenets of spatial assimilation and suggest the persistence of an enduring system of racial stratification" [17]. Upper-income blacks still are segregated and isolated, but not as much as middle- and lower-income blacks.

How segregation got in place and for how long is an important question. Several essential books cover the factors listed above in-depth [21–23]. While the explanation of within-group preferences does play a role, it cannot explain the extent or persistence of segregation. The spatial assimilation hypothesis (essentially social status) may be a factor, but it is inadequate in explaining the levels of racial residential segregation.

Once a neighborhood has become segregated and entrenched, it becomes very difficult to undo or reverse. This is because of the persistence of the reasons presented above. This is despite the extensive efforts to block them nationally, including the Fair Housing Act of 1968 and the Equal Credit Opportunity Act, and private initiatives such as that undertaken by Mapping Prejudice [20,24], as well as the most interesting effort in Hungary to reverse segregation [25]. On top of the discrimination presented above, residents of segregated areas also tend not to want to move very far or even out of their current neighborhood due to friend networks, family ties, and possibly limited knowledge of other neighborhoods [26]. This, in essence, becomes place attachment, and the segregation becomes somewhat more voluntary [25].

## 2. Data and Methods

### 2.1. The Characteristics

The following eight groups were used in this study for both the diversity and segregation calculations. The groups are non-overlapping; thus, each group is exclusive. These follow the definitions used by the U.S. Census [27]: Hispanic (of any race); Non-Hispanic White alone; Non-Hispanic Black alone; Non-Hispanic Native American or Alaska Native alone; Non-Hispanic Asian Alone; Non-Hispanic Native Hawaiian & other pacific Islander alone; Non-Hispanic Some Other race alone; Non-Hispanic Multiracial (two or more races reported).

The data at the block group level was downloaded from the 2020 Decennial Census as CSV files from the U.S. Census Bureau's site [27]. The table from the redistricting data set is P2 "Hispanic or Latino, and not Hispanic or Latino by Race" for the total population and filtered for the specific geographic level. Table P2 contains data for single races alone plus up to 6 racial combinations (a total of 73 variables) [27].

The data files were read into Microsoft Access. An import specification was used to reduce the number of fields to the eight above plus the total population and to convert them to integers. Fields were renamed. The ESRI GIS suite ArcGIS Pro was used to join the variables to the geographic entities.

### 2.2. The Units of Analysis

Two levels of units were used for the analysis. The first unit of analysis is the Metropolitan Area (MA) and Micropolitan Area (MI), as defined by the U.S. Bureau of the Census. MAs and MIs are defined as adjacent whole counties of which the core country contains an Urbanized Area of at least 50,000 people. There may or may not be additional Urban Clusters within the counties. Counties that do not include an urbanized area must be linked to the central county through commuting patterns. The concept is that the Metro and Micro areas represent the economic city. They include physical areas and populations which are not urban. In 2020, there were 927 MAs and MIs excluding Puerto Rico: 384 Metropolitan Areas and 543 Micropolitan Areas. The smallest land area was Vineyard, MA, at 103 square miles, and the largest was the Riverside-San Bernardino-Ontario, CA Metro Area, at 27,300 square miles. The metropolitan and micropolitan area boundaries were downloaded from https://www2.census.gov/geo/tiger/TGRGDB21/tlgdb_2021_a_us_nationgeo.gdb.zip (accessed on 30 April 2022) [28].

For the segregation calculations, the unit of analysis was the census block group. Block group boundaries were obtained from the Census Bureau's nationwide geodatabase: gdb_2021_a_us_substategeo.gdb (accessed on 17 November 2021) [29]. A block group consists of contiguous blocks with a population between 600 and 3000 people. Every census tract has at least one block group but can have up to nine. A block group with the number zero consists entirely of water. Block groups do not cross state, county, or census tract boundaries but can cross city boundaries [27].

There were 242,747 block groups. For this study, several adjustments were made to the block groups. There are 650 block groups that number zero, which means they are water only and have no population. These were removed for analysis. Block groups that otherwise had a total population of zero were removed, leaving a total of 242,097 block groups. Using Arc Pro to perform a spatial join of Metro/Micro boundaries and block groups, those block groups not in a Metropolitan or Micropolitan Area were removed, leaving 221,958 block groups. The median block group size was 0.442 square miles or 283 acres. The smallest block group was in New York City, with only 0.03 acres. The largest was in Alaska, with 14,155 square miles.

### 2.3. Concept Operationalization & Measurement

#### 2.3.1. Diversity

The operational definition of diversity is "... the probability that two units or animals, selected at random, will belong to the same category or species" [30]. Or, stated differently, "... as the probability that randomly paired members of a population will be different on a specified characteristic" [31].

There are several measurements of diversity available [8,31–33]. Nevertheless, the most used diversity index in social and population studies is the Simpson Index, often known as the Diversity Index (DI). The Census Bureau uses this index on its web pages and diversity reports [34–37]. Environmental Systems Research Institute (ESRI) has also used the Simpson Index in its diversity coverage [38].

To be consistent with the bulk of social research, diversity in this report is measured using the Diversity Index (DI).

$$DI = 1 - \sum_{i=1}^{T} p^2$$

where p = the group's proportion of the total; T = the total number of groups

The resulting values range from 0 to 1, where 0 indicates complete homogeneity, and 1 is indicates heterogeneity. The index is often expressed as a percentage to aid interpretation and understandability.

### 2.3.2. Residential Segregation

The task of measuring segregation includes describing the overall extent of the spatial distribution of the population groups. Measuring residential segregation is much more problematic than measuring population diversity, primarily due to space element. Operationally, measuring segregation has been divided into three schools of thought.

As the first step, the school of thought must be established. The schools split on the base of comparison. The first school, the most common, is to define segregation as a relationship of the actual distribution to that of the Lorenz curve, or cumulative percent curve. This school defines perfect integration (zero segregation) as each sub-area containing the same percentage of a minority as the larger area. Segregation is thus the deviation from complete desegregation or equality. Ultimately these are tied to the Gini Index [39,40]. Most studies of segregation use this operational definition and approach to segregation implicitly but not explicitly (Figure 3).

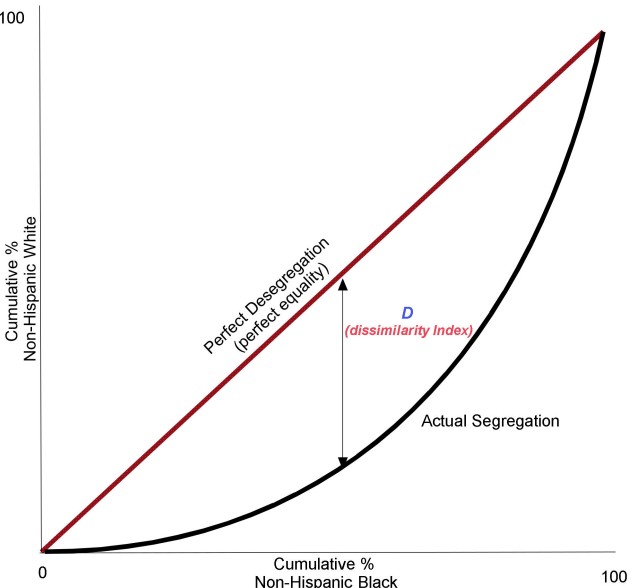

**Figure 3.** The Generalized Segregation Curve.

The second school defines segregation as the deviation of the empirical distribution from what might result from randomness [41,42]. This school defines perfect integration as matching the diffusion created by a random process. Although intriguing, it contains several fundamental problems, but it does help identify clusters. First, the random distribution would be recreated differently every time the data is processed. This would result in more than one random distribution. The second problem is that segregation itself is far from being a random process.

The third school is to make the base of comparison the "expected frequencies" based on marginal frequencies like the expected frequencies for chi-square. This approach by Barrie Morgan [43] and by James Sakoda [44] is for use in multigroup measures. It regards the data as a matrix. This allows a degree of randomness but is much more confined than a completely random approach. This idea provides a repeatable comparison base while still allowing for some degree of randomness. This approach thus defines perfect integration (zero segregation) as each sub-area having the same frequency as its expected frequency. It allows each sub-area to be different. The concept of expected frequencies is derived from marginal totals. By knowing the block group's total population, the metro area's total population, and the proportion of the metro's population in the group, it is possible to

derive the expected number for the group in the block group. The difference this makes is evident in the Figure 4.

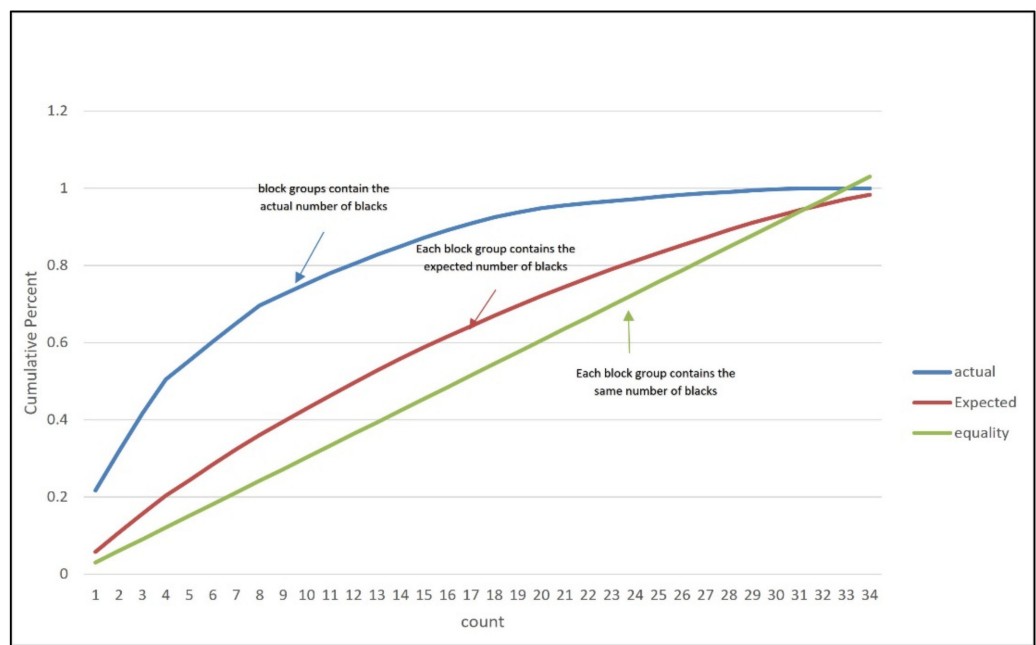

**Figure 4.** Cumulative Percent Curves for Blacks in Aberdeen SD.

The blue curve shows the cumulative percent of the number of blacks found within the block groups. The green line displays the cumulative curve if all block groups contained the same number of blacks. This is the definition of perfect integration for the first school. The red line displays the cumulative curve where each block group contains its expected number of blacks. This is the definition of complete integration in the third school's approach. Notice that using expected frequencies as a comparison base allows for an uneven distribution which is a much more realistic assumption.

Four more matters need to be addressed when measuring residential segregation.

1.   Aspect. The first is the five elements of residential segregation presented first by Massey and Denton [45].

     a.   Evenness. It is defined as the differential distribution of two social groups among the areal units. When the areal units have relatively the same number from each two groups, the greater the evenness—the more uneven the distribution, the greater the segregation.

     b.   Exposure. This refers to the experience of segregation. If the minority and the majority groups share the same neighborhoods, the greater the exposure. There are two aspects, interaction, and isolation, which are measured slightly differently.

     c.   Concentration. It is the physical space that is occupied by a minority in the city.

     d.   Clustering. This is the extent to which a minority group occupies areal units adjacent to each other. High clustering means the presence of one or more ethnic or racial enclaves.

     e.   Centralization. This is the degree to which a group is spatially located near the center of the urban area.

The two authors then compare twenty indices (six for evenness, three for exposure, three for concentration, three for centralization, and five for clustering) on these five aspects [45–47]. Reardon and O'Sullivan conclude that evenness/clustering and exposure/isolation are the most critical dimensions. The other two, centralization and concentration, are subcategories of evenness/clustering [47] (Figure 5).

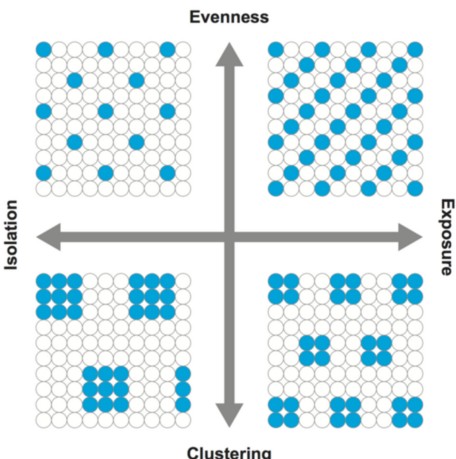

**Figure 5.** The Dimensions of Segregation [12,40].

2. Scale. Three more broad issues with measuring segregation layered over these five dimensions of segregation affect the research direction. The first issue relates to a fundamental methodological conundrum: scale [41,48–50]. "Scale refers to the geographic level at which any phenomenon of interest is organized, experience, or observed… " [51]. The scale in which the analysis is conducted is crucial as it can produce different and sometimes contradictory results [52,53]. Most researchers would agree that in an ideal world, the research would be on an individual or household basis and not a spatially grouped basis. Grigoryeva and Ruef tackled this micro-scale by looking at the actual sequence of the census takers in the 1880 census for Washington D.C. [48]. The pattern of residential areas in the South was for non-whites to live on the same blocks as the whites but back behind them on the alleys. The use of blocks or other areal units shows a level of integration that really was/is not there. This partially accounts for the South showing less segregation than northern cities.

This micro-scale is not feasible as data is not available. Research must proceed with data aggregated by areal units such as census blocks, census tracts, etc. A significant issue is encountered—the Malleable Area Unit Problem (MAUP). This issue then revolves around the problem with any areal unit such as the census tract: The boundaries are arbitrary and can be arranged in many ways. Any larger area can be subdivided into an unlimited number of smaller units.

Since segregation is spatial, measuring it results in an encounter with the Modifiable Areal Unit Problem. Several researchers have produced excellent diagrams illustrating the situations of MAUP. [52]

For the most part, it has been found that the finer the grain, the higher the level of segregation.

3. Pairwise or multigroup. The third concern is how the categories of the population are handled. Almost all dimensions for differentiating a population have more than one or two categories. Measuring diversity does not have this issue. There have been two fundamental approaches to measuring segregation. The first approach with segregation indices is that they typically only deal with two groups at a time and not all the groups, i.e., pairwise versus multigroup. For instance, segregation is measured between blacks and whites, Hispanics, and whites, etc. A multigroup index would consider all the groups (in this study, all eight groups).

Two techniques have been used to address this issue. The first is to compute all the pair-wise combinations and average them together or create a weighted average. The second method is to modify an existing index [51] or create a new one [44,54].

4. Proximity. The fourth concern is how proximity and how space itself is managed, that is, how to account for the relationship between the areal units. In many ways, these are

an attempt to overcome issues of the MAUP: aspatially, that is, the measurements are an overall view or a global view, disregarding spatial proximity, or the measurement deals with the issue of proximity (local view) [12,54].

Two approaches have been proposed. One is to take a traditional global index and adjust it to account for spatial dimensions. An example is the Spatial dissimilarity Index (SD) discussed by Oka and Wong [53], which factors in the population of adjacent areal units. Other techniques try to factor in the actual distance between the groups (or between the centroids of the areal units), such as the Spatial Proximity Index (P) proposed by White [14]. With the advances in spatial statistics and Geographic Information Systems (GIS), technics have become more robust.

These spatially adjusted indexes of segregation are still global. That is, the result is an overall value summarizing the area.

Numerous spatial statistics directly consider distance that could be used, and they are good indicators of clustering, such as Moran's I, Geary's c, and Getis-Ord G. These also have local indicators of spatial association (LISA) techniques that can pinpoint the clusters and be mapped.

In this report, three measures of segregation are used. The first one is the Index of Dissimilarity (D). Operationally, segregation is the amount of deviation from complete desegregation or how uniform the distribution is, where each analysis unit contains the same proportions as the larger unit. This index is a non-spatial pair-wise index of evenness/clustering. The equation for the Dissimilarity Index is:

$$D = \sum_{i=1}^{n} \frac{1}{2} \left| \frac{b_i}{B} - \frac{w_i}{W} \right|$$

where $n$ = the number of subareas (block groups, BG); $b_i$ = the number non-Hispanic blacks in $BG_i$; $w_i$ = the number of non-Hispanic whites in $BG_i$; B = the total number of non-Hispanic blacks in the Metro Area; W = the total number of non-Hispanic whites in the Metro Area.

The second index used is the aspatial Isolation Index (P*). This index is a measure of the exposure dimension of segregation. It gives the probability of interracial interaction within the geographic area. The equation is:

$$P_b^* = \sum_{i=1}^{n} \left( \frac{b_i}{B} \right) \left( \frac{b_i}{T} \right)$$

where $n$ = the number of subareas (block groups, BG); $b_i$ = the number non-Hispanic blacks in $BG_i$; B = the total number of non-Hispanic blacks in the Metro Area; T = the total population of the Metro Area.

This amounts to the weighted average of each block group's proportion of non-Hispanic blacks. It can be calculated for each of the eight racial categories. Interpreting the index as configured, it provides the probability of intra-group interaction. The index runs from zero to 1 (or 100%). The lower the value, the less intra-group interaction and the greater the inter-group interaction. A higher value indicates a greater probability of intra-group interaction, and a lower value indicates the inter-group interaction. Quoting Lieberson, Robinson defines the Isolation Index operationally as "... 'average probability of interacting with some specified population based on the distribution of persons by subareas and the assumption that interaction is with someone in the same subarea'" [55].

The third indicator used is the Multigroup Dissimilarity Index (MG-D or $D_G$), initially proposed by James Sakoda in 1981 [44]. This uses expected frequencies, as discussed above. The use of expected frequencies as the base will produce different results than when using

the traditional base. This researcher cannot find studies that have used this approach before. The equation for this is:

$$D_G = \frac{1}{2} \frac{\sum_j \sum_i \left| N_{ij} - E_{ij} \right|}{\sum_j N P_j (1 - P_j)}$$

$$\text{and } E_{ij} = \frac{N_i \; N_j}{N}$$

where $N_{ij}$ = the population racial group j of block group i; N = the total population of the Metro Area; $E_{ij}$ = expected number of racial group j in BG i; $P_{ij}$ = the proportion of racial group j of BG i population; $P_j$ = the proportion of racial group j of the metro area; $D_j$ = the proportion of racial group j of the Metro Area population

### 2.4. The Hypotheses

The research hypothesis is:

**Hr.** *The greater the degree of population racial diversity, the greater the level of residential racial segregation.*

Since there are three indicators of residential segregation used, there are three operational hypotheses:

**H1.** *Within Metropolitan and Micropolitan Areas and based upon block groups, the greater the degree of population differentiation on the characteristic of race as measured by the Diversity Index (DI), the greater the level of overall residential racial segregation as measured by the Multigroup Dissimilarity Index ($D_G$).*

**H1a.** *The nature of the relationship is positive and linear; an increase in diversity increases the Index of Multigroup Dissimilarity.*

**H2.** *Within Metropolitan and Micropolitan Areas and based upon block groups, the greater the degree of population differentiation on the characteristic of race as measured by the Diversity Index (DI), the greater the level of residential racial segregation as measured by the Index of Dissimilarity (D).*

**H2a.** *The nature of the relationship is positive and linear; an increase in diversity increases the Index of Dissimilarity.*

**H3.** *Within Metropolitan and Micropolitan Areas and based upon block groups, the greater the degree of population differentiation on the characteristic of race as measured by the Diversity Index (DI), the greater the level of residential isolation, as measured by the Isolation Index, (P\*).*

**H3a.** *The nature of the relationship is positive and linear; an increase in diversity increases the isolation Index.*

There are three null hypotheses:

**H1-0.** *Any relationship between the degree of urban racial diversity and the degree of urban multigroup residential racial segregation is due to chance or is random.*

**H2-0.** *Any relationship between the degree of urban racial diversity and the degree of urban residential racial segregation is due to chance or is random.*

**H3-0.** *Any relationship between the degree of urban racial diversity and the degree of urban residential racial isolation is due to chance or is random.*

While the hypotheses propose linear relationships, the relationships are probably not linear. As seen in Figure 6, a positive relationship can take many different forms, of which being linear is only one.

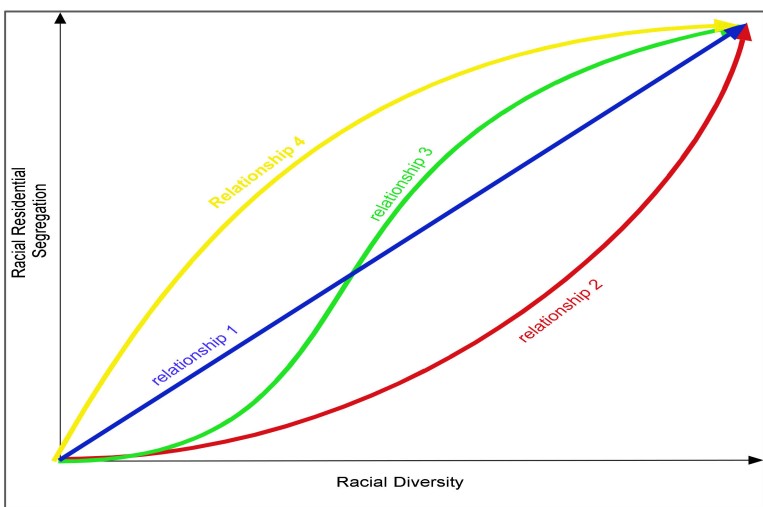

**Figure 6.** Generalized possible Relationships.

The idea that a threshold (size of the minority group) triggers greater segregation appears in much research on segregation. The smallest level is at the neighborhood level. Here, it is often termed "the tipping point." There are three components to the tipping point: the preference point, the leaving point, and the willingness-to-enter point [18]. When a given neighborhood receives a certain percentage of blacks (or Hispanics, Asians, or any no-dominant racial or ethnic group), the dominant group, usually non-Hispanic whites, moves out. While this percentage is somewhat indefinite, research has found it to be twenty-five percent or lower [17]. Researchers suggest that increases in the proportion of blacks (or other racial groups) increase residential segregation at the metropolitan or city level. This may be hard to find at high levels of aggregation. The Figure 7 shows possible relationships between the proportion of black in the metro/micro areas and the Dissimilarity Index for blacks and whites.

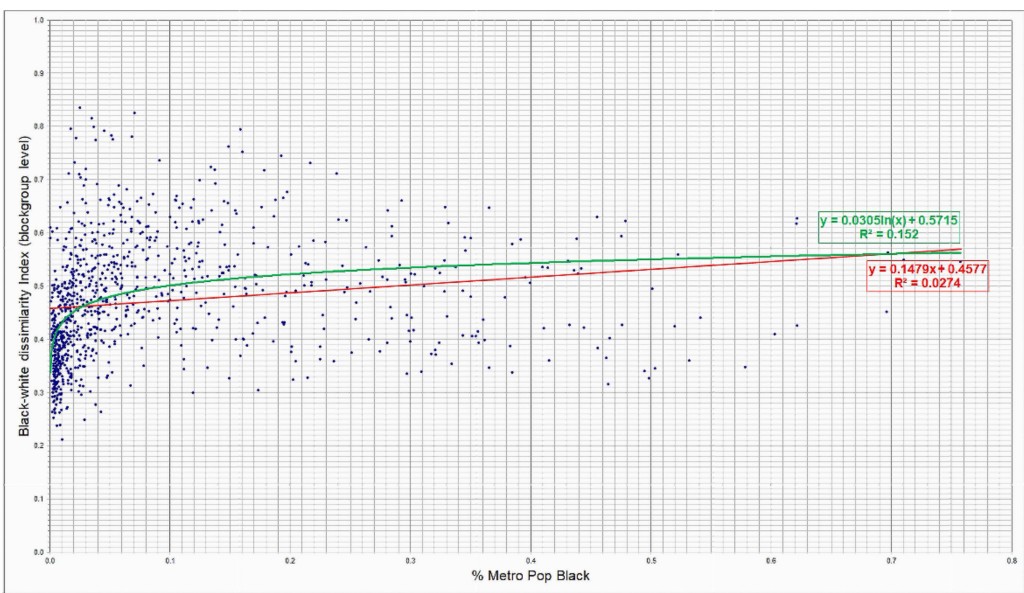

**Figure 7.** The Relationship Between the percent Black and Black-White Dissimilarity Index.

Earlier research has argued on both sides of the hypothesis. One of the few studies to address the connection was from Lee, Iceland, and Farrell. These authors point out that "Increasing diversification in metropolitan and micropolitan areas across the country does not necessarily mean that people of different ethnoracial groups are now more apt to share neighborhoods" [56]. Their unstated hypothesis is for a negative relationship where an

increase in racial diversity will decrease residential segregation. Increasing population diversity may not result in increased residential integration, that is, a decrease in residential segregation. This, in essence, is the converse of the hypothesis of this study.

The authors used Thiel's H, the multigroup information theory index, supplemented with the Dissimilarity Index (D). Using H, they conclude:

> *"We find that H has declined substantially in metropolitan areas, from 34 in 1980 to 23 in 2010 (Figure 13.5). That is, metro residents now live in census tracts that, on average, are 23 percent less diverse (or more segregated) than the metropolis as a whole, down from 34 percent less diverse three decades earlier."*

This conclusion contains contradictory statements. The first is the implication of H declining, that segregation overall had declined. That was followed by stating that there was an increase in residential segregation as the neighborhoods were less diverse. Following up using the Dissimilarity Index (D), they report that "Blacks . . . represented the most segregated group in each year; at the same time, they experienced the greatest decreases over the three-decade period" [56]. They then suggest that this was the result of the increased Hispanic population. Their general conclusion was: ". . . growing ethnoracial diversity across metropolitan and micropolitan America does not go hand in hand with consistently high (or rising) levels of neighborhood segregation, as anticipated by the ethnic stratification perspective" [56]. Herein lies their unstated hypotheses of a negative relationship where an increase in racial diversity will decrease residential segregation.

Logan and Stults using the Dissimilarity Index (D) and the Exposure/Isolation Index (P*), concur that the increase in the neighborhood diversity of whites and blacks is due to the increasing number of Hispanics and Asians:

> *"The trend is clearly toward increasing diversity for whites and blacks in their neighborhoods because of the growing share of Hispanics and Asians in the overall population. The average white person now lives in a neighborhood with considerably larger shares of Hispanics and Asians, but only small increases of African Americans since 1980. African Americans now have more Hispanic and Asian neighbors, as well as a small increase in co-residence with whites."* [57]

Logan and Stults continue adding the following points.

> *". . . whites rarely move into minority neighborhoods. Formerly all white neighborhoods are becoming more diverse as new groups move into them. There are many cases like this, but they are countered by growing segregation between other neighborhoods."*

> *"Hispanics and Asians have been moving toward new destinations since the 1980s, and this represents movement toward areas where they are less segregated. Yet in the process, their arrival has been met with increasing segregation."*

Examining trends since the '40s, they report that "There has been almost no change in the share of white neighbors for the average African American in this whole period" [56]. They found that the isolation of blacks, Hispanics, and Asians, for the most part, has increased. Concluding for blacks: "What is most striking about these figures is that with very few exceptions, the Isolation Index is above 40 in the largest metro regions. African Americans live in neighborhoods where they are an absolute majority, or a near majority, in most of these places" [57].

Logan and Stults also report that both blacks and Hispanics were more segregated and isolated in Metro Areas where either of these groups had large shares of the metro population. They found that the metro areas where Hispanics and Asians moved to over the decades increased segregation and isolation while increasing diversity [52].

Nat Silver reinforces Logan and Stults in discussing Chicago:

> *"Chicago deserves its reputation as a segregated city. But it is also an extremely diverse city. And the difference between those terms—which are often misused and misunderstood—says a lot about how millions of American city dwellers live. It is*

*all too common to live in a city with a wide variety of ethnic and racial groups—including Chicago, New York, and Baltimore—and yet remain isolated from those groups in a racially homogenous neighborhood."* [58]

In a study in Denmark, Dinesen and Sonderskov found evidence that ethnic diversity in a micro-context negatively affects social trust due to dispositional skepticism towards others of a different ethnic background. The effects were ethnic group isolation and segregation [59].

There is some support for the hypotheses that racial segregation and racial isolation have been increasing along with increasing racial diversity.

## 3. Results

### 3.1. Diversity

There is little disagreement that we as a nation are very racially diverse. For all 927 metro and micro areas, the median Diversity Index (DI) was 0.399 (the mean was 0.397). There was a 40% chance that two people picked randomly would be of a different race. There was a wide range of indices, running from 0.045 to 0.777. Twenty-five percent of the areas had an index of 0.260 or below, while twenty-five percent had an index of 0.403 or above. The distribution of the metro/micro areas was not normal but was bimodal and would be more of a sine wave (Figure 8).

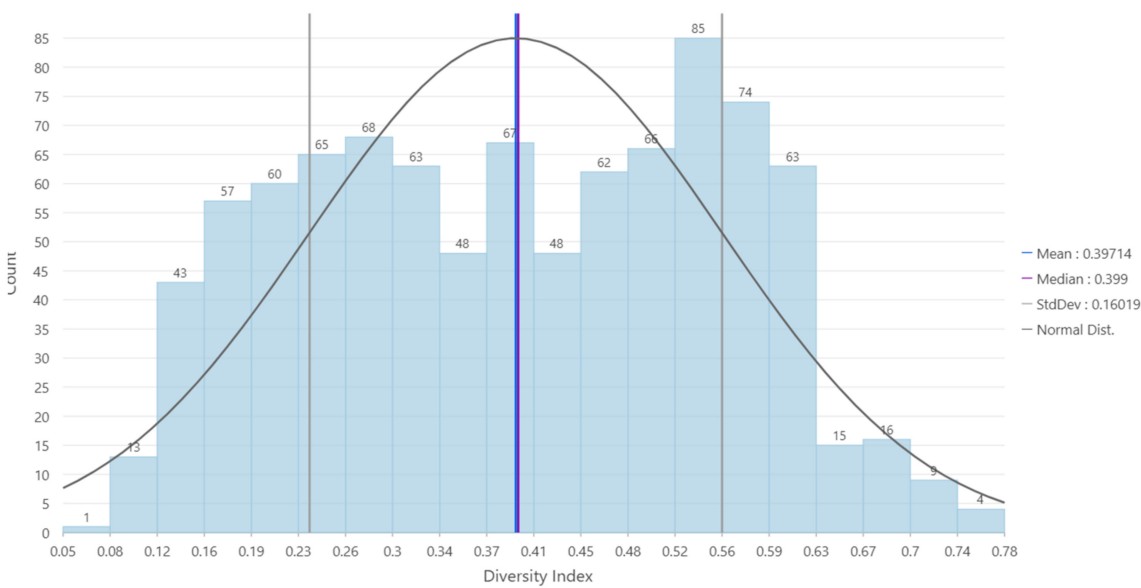

**Figure 8.** Histogram of Racial diversity Indices, Metro/Micro Areas, 2020.

The 384 metropolitan areas (MA) with a median Diversity Index of 0.471 were more diverse than the 543 micropolitan areas (MI), with an index of 0.334. Thus, two people chosen randomly in a metropolitan area were more likely to be of a different race (47%) than two people chosen at random in a micropolitan area (33%).

Four of the ten most diverse metro/micro areas were found in Hawaii, and some of the larger metro areas were found in the ten most diverse. The Hilo Micro Area has the highest Diversity Index at 0.7768; two people chosen at random would have 77.68% of being of a different race. The Honolulu Metro Area was fifth with an index of 0.7358. San Francisco was seventh, Las Vegas was eighth, and Washington DC was ninth with an index of 0.7194.

In contrast, are the metro/micro areas that had the least diversity, i.e., they were more homogenous. Three of these were in Texas. The Rio Grande City Micro Area was the least diverse, with an index of 0.0455. Two people were chosen randomly here, only had a change of 4.55% of being of a different race. In tenth was Warren, Pennsylvania, with an index of 0.1130 (Tables 1 and 2).

**Table 1.** Diversity Index: Ten Highest Metro/Micro Areas, 2020.

| Metro/Micro Area | Total Population | Population Density | DI |
|---|---|---|---|
| Hilo, HI Micro Area | 200,629 | 49.80 | 77.68% |
| Kahului-Wailuku-Lahaina, HI Metro Area | 164,754 | 141.84 | 77.14% |
| Kapaa, HI Micro Area | 73,298 | 118.245 | 76.59% |
| Vallejo, CA Metro Area | 453,491 | 551.807 | 75.64% |
| Urban Honolulu, HI Metro Area | 1,016,508 | 1692.42 | 73.58% |
| Lumberton, NC Micro Area | 116,530 | 123.013 | 73.55% |
| San Francisco-Oakland-Berkeley, CA Metro Area | 4,749,008 | 1922.40 | 73.51% |
| Las Vegas-Henderson-Paradise, NV Metro Area | 2,265,461 | 287.07 | 72.05% |
| Washington-Arlington-Alexandria, DC-VA-MD-WV Metro Area | 6,385,162 | 972.206 | 71.94% |
| Trenton-Princeton, NJ Metro Area | 387,340 | 1725.51 | 71.20% |

**Table 2.** Diversity Indexes: Ten Lowest Metro/Micro Areas, 2020.

| Metro/Micro Area | Total Population | Population Density | DI |
|---|---|---|---|
| Rio Grande City-Roma, TX Micro Area | 65,920 | 53.89 | 4.55% |
| Laredo, TX Metro Area | 267,114 | 79.46 | 9.20% |
| St. Marys, PA Micro Area | 30,990 | 37.49 | 9.48% |
| Eagle Pass, TX Micro Area | 57,887 | 45.24 | 9.84% |
| Mount Gay-Shamrock, WV Micro Area | 32,567 | 71.78 | 10.54% |
| Coshocton, OH Micro Area | 36,612 | 64.92 | 10.68% |
| Jackson, OH Micro Area | 32,653 | 77.69 | 10.70% |
| Greenville, OH Micro Area | 51,881 | 86.74 | 10.72% |
| Spirit Lake, IA Micro Area | 17,703 | 46.52 | 10.81% |
| Warren, PA Micro Area | 38,587 | 43.64 | 11.30% |

Geographically, the diversity indexes displayed distinct regional patterns. The metro/micro areas along the East, Gulf, West coasts, and in the South are much more racially diverse than the areas in the "rust belt" and the states surrounding the great lakes. Those areas adjacent to the Mexican border were the exception to this pattern (Figure 9).

The Diversity Indices by block group produced an interesting distribution. As seen in Figure 10, the distribution rises rather sharply, flattens out, peaks just short of the plus one standard deviation, and drops off sharply. The median index for block groups was 0.439, and the mean was 0.424. The indexes ranged between zero and 0.854. Thus, some block groups were completely homogenous racially. Twenty-five percent had an index of 0.262 or less, and twenty-five percent had an index of 0.586 and above. The block groups followed the same pattern as the metro and micro areas. Block groups in the micro areas (23,154) portrayed lower diversity, with a median of 0.263 than metro areas (198,066), with a median Diversity Index of 0.458. The most diverse block group was found in Anchorage, Alaska (Figure 10).

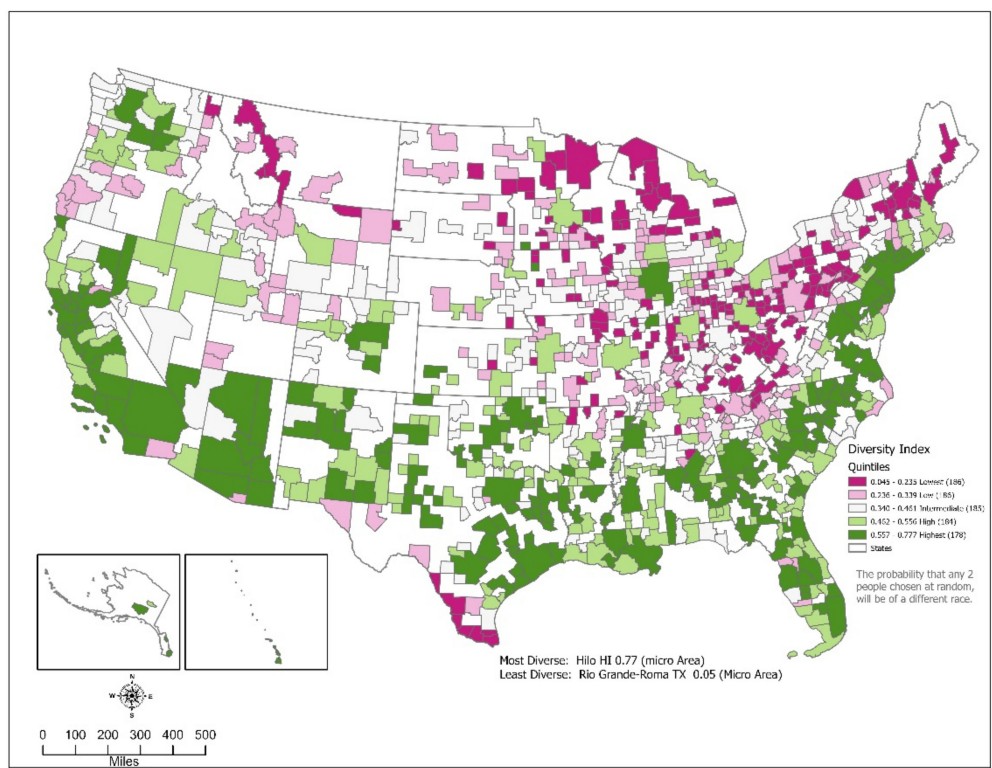

**Figure 9.** Metropolitan and Micropolitan Area Racial Diversity Indexes, 2020.

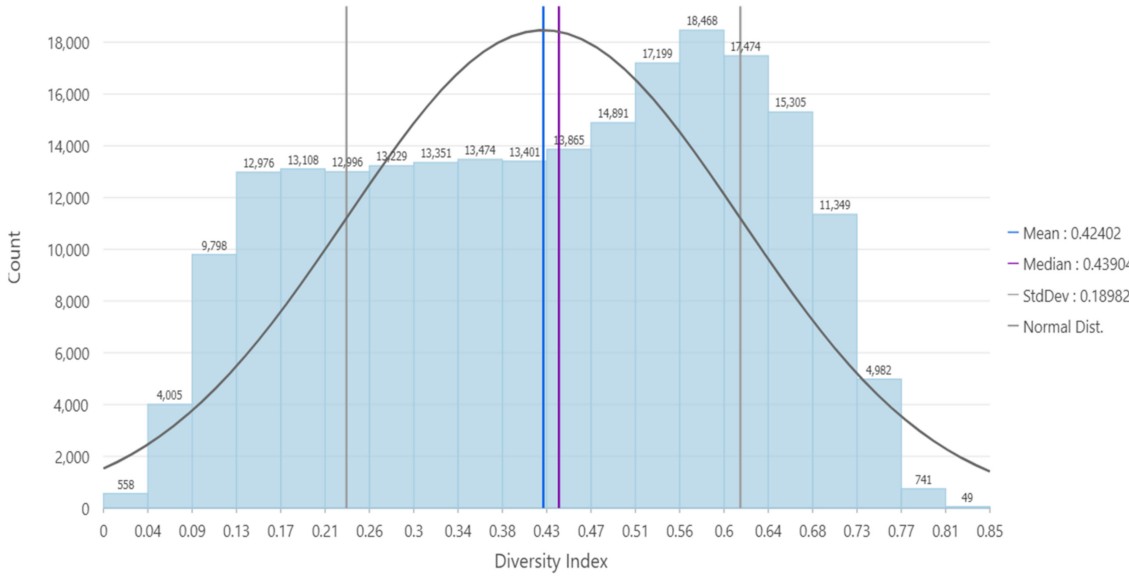

**Figure 10.** Histogram of Block Group Diversity Indices, 2020.

### 3.2. Segregation

3.2.1. Multigroup Dissimilarity Index ($D_G$)

The Multigroup dissimilarity Index ($D_G$) measures spatial evenness for all eight racial groups across a metro/micro area. All metro and micro areas had a median Multigroup Dissimilarity Index of 0.318, and the mean index was 0.323. This index is interpreted in the same manner as the pair-wise dissimilarity index: just under 32% of the total population in all 927 areas would have to move to obtain integration, defined as the expected distribution. Twenty-five percent of the areas had a $D_G$ of 0.256 or less. Moreover, twenty-five percent had an index of 0.387 or more. The maximum index was 0.690, and the lowest was 0.117.

The distribution of $D_G$ is remarkably close to a normal distribution. It had a kurtosis value of 2.8, not particularly peaked or flat. The distribution had a positive skew. It had a slight tail to the right towards the higher index values (Figure 11).

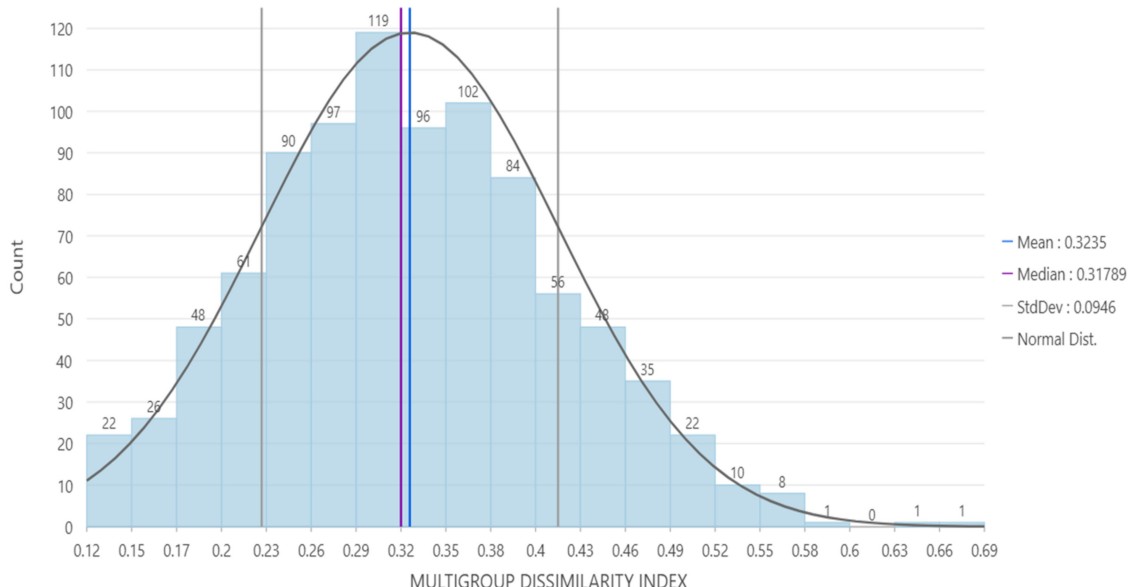

**Figure 11.** Histogram of Metro/Micro Area Multigroup Dissimilarity Indices, 2020.

The 384 metropolitan Areas with a median $D_G$ of 0.362 were more segregated than the 543 micropolitan areas, with a median index of 0.293. The micro areas showed a greater variability of indexes with a range of 0.573 than the metro areas, whose range was 0.441.

The metro/micro area with the highest Multigroup Dissimilarity Index was Show Low, Arizona, with an index of 0.690. Here, 69% of the total population would have to move from their current block group to achieve the expected level of residential integration. Three of the highest ten were in the South: Greenwood, Mississippi; Cleveland, Mississippi; and Monroe, Louisiana. Three of the areas had an index of just over 0.55 and a population of over a million: Cleveland, Ohio; Milwaukee, Wisconsin; and Detroit, Michigan. In each of these three areas, over $\frac{1}{2}$ million people would have to move to achieve integration of all racial groups.

The metro/micro area with the lowest Multigroup Dissimilarity Index was Sandpoint, Idaho, with an index of 0.117. Only eleven percent of its total population would need to move to obtain the expected level of residential integration. Coeur d'Alene Idaho followed it with an index of 0.118 which also had the largest population (171,362) of this group of ten (Tables 3 and 4).

**Table 3.** Ten Highest Metro/Micro Areas Multigroup Dissimilarity Indices, 2020.

| Metro/Micro Area | Total Population | Population density | DI | $D_G$ |
|---|---|---|---|---|
| Show Low, AZ Micro Area | 106,717 | 10.73 | 0.622 | 0.690 |
| Malone, NY Micro Area | 47,555 | 29.19 | 0.332 | 0.634 |
| Lexington, NE Micro Area | 26,004 | 17.68 | 0.540 | 0.588 |
| Greenwood, MS Micro Area | 38,337 | 31.36 | 0.506 | 0.564 |
| Nogales, AZ Micro Area | 47,669 | 38.56 | 0.286 | 0.561 |

**Table 3.** *Cont.*

| Metro/Micro Area | Total Population | Population density | DI | D$_G$ |
|---|---|---|---|---|
| Cleveland-Elyria, OH Metro Area | 2,088,251 | 1044.76 | 0.502 | 0.559 |
| Milwaukee-Waukesha, WI Metro Area | 1,574,731 | 1082.39 | 0.547 | 0.558 |
| Cleveland, MS Micro Area | 30,985 | 35.35 | 0.510 | 0.558 |
| Detroit-Warren-Dearborn, MI Metro Area | 4,392,041 | 1128.40 | 0.540 | 0.551 |
| Monroe, LA Metro Area | 207,104 | 90.75 | 0.561 | 0.551 |

**Table 4.** Ten Lowest Metro/Micro Areas Multigroup dissimilarity Indices, 2020.

| Metro/Micro Area | Total Population | Population Density | DI | D$_G$ |
|---|---|---|---|---|
| Sandpoint, ID Micro Area | 47,110 | 27.181 | 0.189 | 0.117 |
| Coeur d'Alene, ID Metro Area | 171,362 | 138.443 | 0.237 | 0.118 |
| Durant, OK Micro Area | 46,067 | 50.945 | 0.536 | 0.119 |
| Roseburg, OR Micro Area | 111,201 | 22.083 | 0.285 | 0.128 |
| Los Alamos, NM Micro Area | 19,419 | 177.955 | 0.483 | 0.128 |
| West Plains, MO Micro Area | 39,750 | 42.872 | 0.178 | 0.128 |
| Grants Pass, OR Metro Area | 88,090 | 53.758 | 0.308 | 0.128 |
| Tahlequah, OK Micro Area | 47,078 | 62.844 | 0.678 | 0.131 |
| Spearfish, SD Micro Area | 25,768 | 32.208 | 0.204 | 0.132 |
| Harrison, AR Micro Area | 44,598 | 31.612 | 0.176 | 0.132 |

Geographically, the distribution of metro/micro areas on their Multigroup Dissimilarity Indices matches the diversity Indices. Metro/micro areas having the highest Multigroup Dissimilarity Indices are overly represented in the East, Great Lakes region, and the South. The Rust Belt and the Northeast are of interest, where the metro/micro areas had low diversity indices and high Multigroup Dissimilarity Indices (Figure 12).

### 3.2.2. Pair-Wise Dissimilarity Index

The Multigroup Dissimilarity Index hides details and spatial variations about individual racial groups. Blacks were the racial group that was the most segregated. The median pair-wise Dissimilarity Index for blacks and whites was 0.462 (the mean was 0.471). Forty-six percent of the blacks would have to move to different block groups within the Metro/Micro Areas to obtain complete residential integration. The lowest index was 0.211, and the highest was 0.835. The lowest metro/micro white-Black dissimilarity Index indicates that a minimum of 21 percent of the blacks would have to move, while in the most segregated area, 84 percent of the blacks would have to move from their current block group to another one to achieve residential integration.

Black-white segregation was located heavily in the eastern half of the country (Figure 13), where most were in the upper quintile of areas with a Black-White Dissimilarity Index of 0.560 or greater. Fifty-six percent of the blacks living in these areas would have to move to another block group for the areas to achieve perfect residential integration. The most segregated metro/micro area was Somerset, Pennsylvania, with an index of 0.835; eighty-four percent of the blacks living here would have to move. The largest metropolitan area in the highest ten was Milwaukee, Wisconsin, with an index of 0.794.

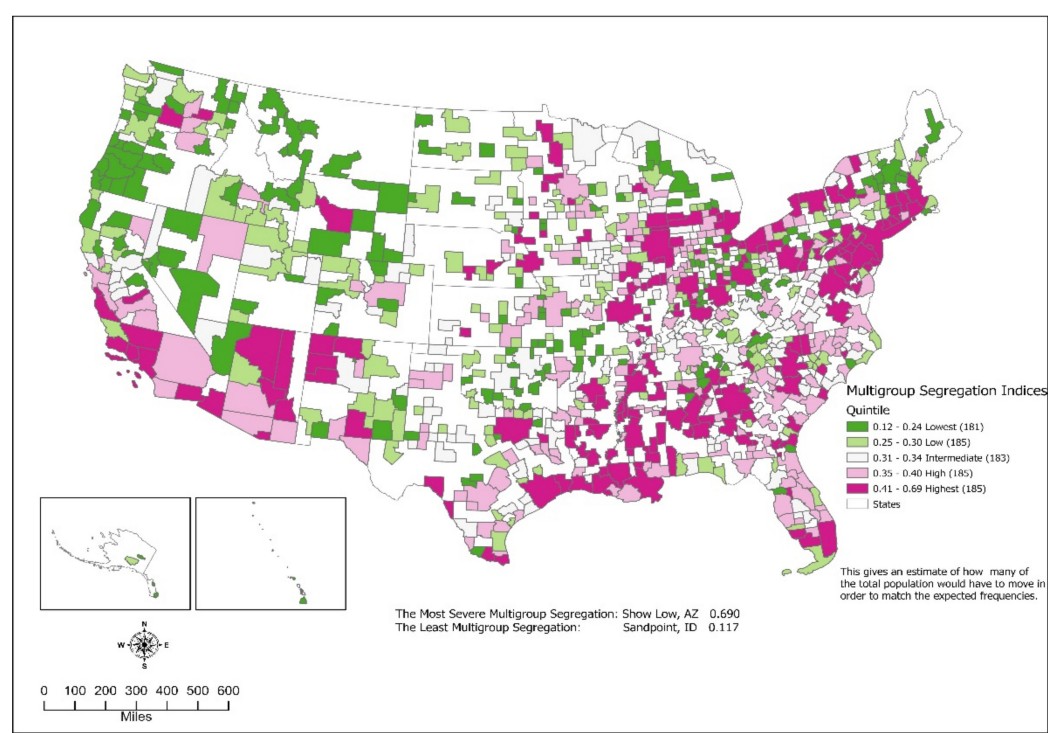

**Figure 12.** Multigroup Dissimilarity Indices, Metro/Micro Areas 2020.

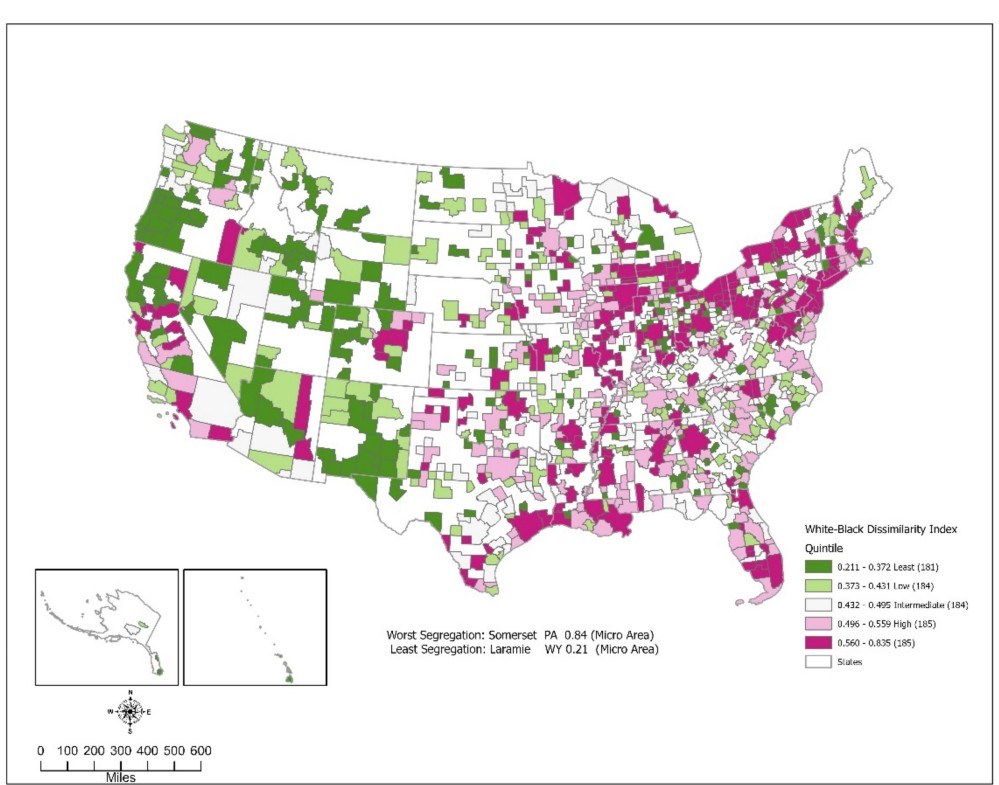

**Figure 13.** White-black Segregation Indices, Metro/Micro Areas 2020.

Even at the lowest indices, over twenty percent of the blacks would have to move to achieve perfect residential integration. In contrast, the metro/micro area with the lowest index was Laramie, Wyoming, with an index of 0.211. Spenser, Iowa followed with an index of 0.237.

Hispanics were less segregated from whites than blacks were, with a median of 0.312 Hispanic-White Dissimilarity Index, and the mean was 0.327. The lowest index was

0.108, while the highest was 0.663. The first quartile (twenty-five percent of the metro/micro areas were under this) was 0.255. The third quartile index was 0.389 (twenty-five percent of the areas were over this value). Metropolitan Areas were more segregated than the Micropolitan Areas, having a Median index of 0.352 versus 0.292. The maximum index was a micro area with an index of 0.663.

Hispanics display a similar distribution as for Black-white segregation. Hispanic-White segregation was located heavily in the eastern half of the country (Figure 14). However, the areas in the Southwest and California register high indices. Most of the metro/micro areas in the country's eastern half have a Hispanic-White dissimilarity Index of 0.441 or greater (the upper quintile of areas). Forty-one percent of the Hispanics living in these metro/micro areas would have to move to another block group for the areas to achieve perfect residential integration.

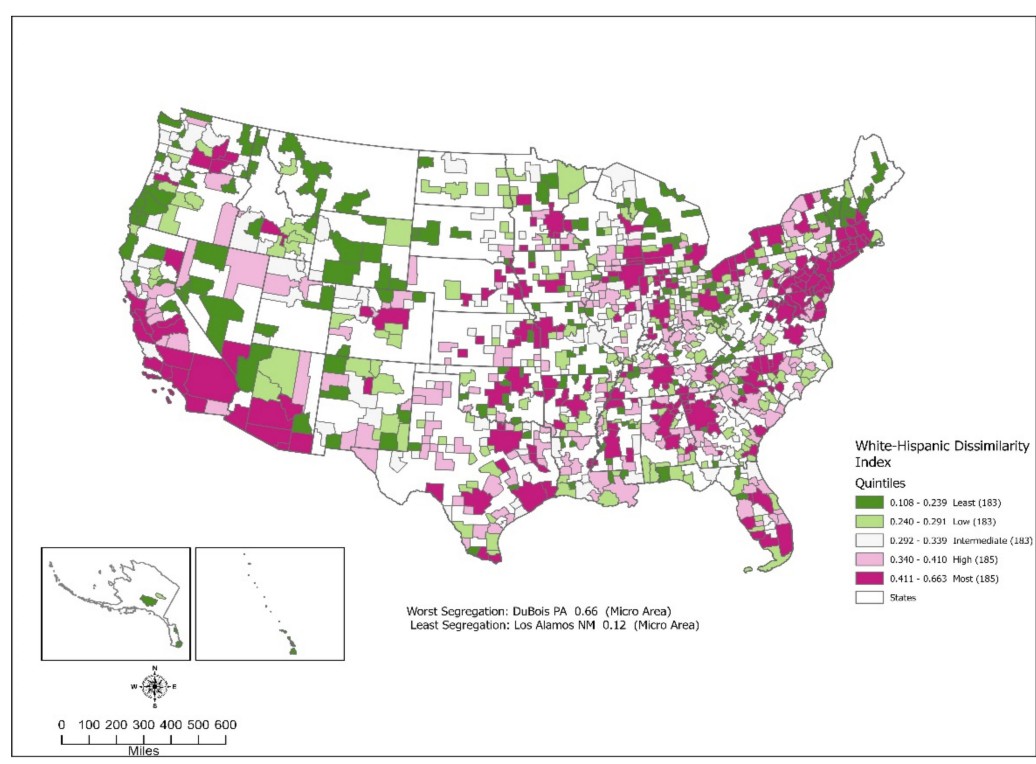

**Figure 14.** Hispanic-White Dissimilarity Indices, Metro/Micro Areas, 2020.

The metro/micro area registering the highest index was DuBois, Pennsylvania, with a Hispanic-White dissimilarity Index of 0.663. The second highest was Lexington, Nebraska, with an index of 0.651. The metro/micro areas with the lowest indices were led by Los Alamos, New Mexico, with an index of 0.108. Fallon, Nevada, was next at 0.110.

Native American-White segregation was low, with a median index of 0.380 and a mean index of 0.391. The lowest index was 0.091 (the lowest of any of the racial groups), but the highest index was 0.891 (the highest of any racial group); thus, Native American-White segregation displayed the widest range of indices. Twenty-five percent of areas had an index of 0.314 or less, while twenty-five percent had an index of 0.458 or greater. In many ways, the metro and micro areas had similar distributions; the median was 0.388 versus 0.373, respectively. The Micro areas displayed the broadest range of indices, containing both the lowest and highest indexes.

Not surprisingly, the geographic distribution of Native American-White dissimilarity indices displays great variation (Figure 15). Segregation was highest in the Mid-Atlantic/Northeast, the upper Midwest, and the Southwest. The pattern in the West has been created by the historical treatment of Native Americans being forced, often by gunpoint, into reservations with few, if any, whites. Oklahoma is an exception. The state has

nine of the ten lowest areas on the Native American-White dissimilarity Index. Durant, Oklahoma's index was 0.091. The next lowest was Miami, Oklahoma, at 0.099.

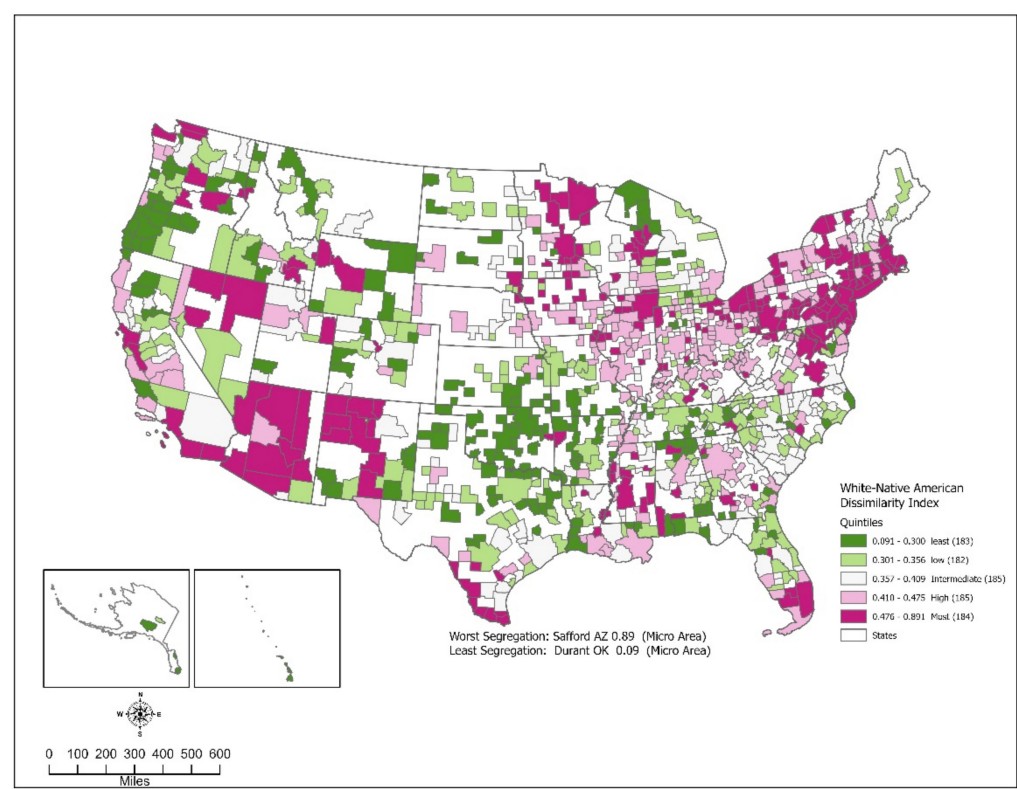

**Figure 15.** Native American-White Dissimilarity Indices, Metro/Micro Areas, 2020.

Asian-White segregation was in the intermediate range with a median index of 0.399 and a mean index of 0.402. The lowest index was 0.1391, but the highest index was 0.729; thus, Asian-White segregation displayed a range of 0.593. Twenty-five percent of the areas had an index of 0.341 or less, while twenty-five percent had an index of 0.467 or greater. In many ways, the metro areas were more segregated than the micro areas; the median was 0.427 versus 0.385, respectively. The Micro areas displayed a broader range of indices than the metro areas (Tables 5 and 6).

**Table 5.** Ten Highest Metro/Micro Areas for Black-White, Hispanic-White, Native American-White, and Asian-White Dissimilarity Indices 2020.

| Black-White Dissimilarity Index | | Hispanic-White Dissimilarity Index | | Native American-White Dissimilarity Index | | Asian-White Dissimilarity Index | |
|---|---|---|---|---|---|---|---|
| Metro/Micro Area | Index | Metro/Micro Area | Index | Metro/Micro Area | Index | Metro/Micro Area | Index |
| Somerset, PA Micro | 0.835 | DuBois, PA Micro | 0.663 | Safford, AZ Micro | 0.891 | Pearsall, TX Micro | 0.729 |
| Sault Ste. Marie, MI Micro | 0.825 | Lexington, NE Micro | 0.651 | Malone, NY Micro | 0.872 | Utica-Rome, NY Metro | 0.674 |
| Malone, NY Micro | 0.815 | Reading, PA Metro | 0.644 | Show Low, AZ Micro | 0.862 | Napa, CA Metro | 0.662 |
| Cañon City, CO Micro | 0.799 | Salinas, CA Metro | 0.629 | EspaÃ±ola, NM Micro | 0.833 | Lafayette-West Lafayette, IN Metro | 0.638 |
| Sonora, CA Micro | 0.795 | Los Angeles, CA Metro | 0.609 | Alamogordo, NM Micro | 0.831 | Cordele, GA Micro | 0.630 |
| Milwaukee, WI Metro | 0.794 | Springfield, MA Metro | 0.605 | Payson, AZ Micro | 0.828 | Rio Grande City-Roma, TX Micro | 0.633 |
| Lexington, NE Micro | 0.792 | Kendallville, IN Micro | 0.603 | Eagle Pass, TX Micro | 0.82 | Cortland, NY Micro | 0.630 |
| Huntingdon, PA Micro | 0.783 | Nogales, AZ Micro | 0.596 | Roanoke Rapids, NC Micro | 0.813 | Storm Lake, IA Micro | 0.628 |
| Susanville, CA Micro | 0.781 | New York City, Metro | 0.593 | Meridian, MS Micro | 0.802 | Bay City, TX Micro | 0.624 |
| DuBois, PA Micro | 0.778 | Trenton-Princeton, NJ Metro | 0.585 | Rio Grande City-Roma, TX Micro | 0.797 | Blacksburg-Christiansburg, VA Metro | 0.623 |

**Table 6.** Ten Lowest Metro/Micro Areas for Black-White, Hispanic-White, Native American-White, and Asian-White Dissimilarity Indices, 2020.

| Black-White Dissimilarity Index | | Hispanic-White Dissimilarity Index | | Native American-White Dissimilarity Index | | Asian-White Dissimilarity Index | |
|---|---|---|---|---|---|---|---|
| **Metro/Micro Area** | **Index** | **Metro/Micro Area** | **Index** | **Metro/Micro Area** | **Index** | **Metro/Micro Area** | **Index** |
| Laramie, WY Micro | 0.211 | Los Alamos, NM Micro | 0.108 | Durant, OK Micro | 0.091 | Shelton, WA Micro | 0.136 |
| Spencer, IA Micro | 0.237 | Fallon, NV Micro | 0.110 | Miami, OK Micro | 0.099 | Winnemucca, NV Micro | 0.167 |
| Casper, WY Metro | 0.238 | Coeur d'Alene, ID Metro | 0.117 | Muskogee, OK Micro | 0.114 | Truckee-Grass Valley, CA Micro | 0.168 |
| Pahrump, NV Micro | 0.249 | Kalispell, MT Micro | 0.132 | Ada, OK Micro | 0.118 | Hood River, OR Micro | 0.175 |
| Brookings, OR Micro | 0.253 | Sandpoint, ID Micro | 0.133 | McAlester, OK Micro | 0.127 | Brookings, OR Micro | 0.178 |
| Glenwood Springs, CO Micro | 0.26 | Pahrump, NV Micro | 0.141 | Tahlequah, OK Micro | 0.131 | Vineyard Haven, MA Micro | 0.179 |
| The Dalles, OR Micro | 0.262 | Winnemucca, NV Micro | 0.148 | Bartlesville, OK Micro | 0.147 | Pahrump, NV Micro | 0.187 |
| Pella, IA Micro | 0.262 | Price, UT Micro | 0.149 | Duncan, OK Micro | 0.151 | Carson City, NV Metro | 0.187 |
| Vernal, UT Micro | 0.264 | Vineyard Haven, MA Micro | 0.150 | Ardmore, OK Micro | 0.157 | Spearfish, SD Micro | 0.190 |
| Lawrence, KS Metro | 0.264 | Menomonie, WI Micro | 0.154 | Escanaba, MI Micro | 0.170 | Fremont, NE Micro | 0.197 |

The geographic distribution of Asian-White dissimilarity indices displays a great deal of variation (Figure 16). Segregation was highest in the Mid-Atlantic/Northeast, Midwest, and the South. California had several areas with high segregation. The Mountain region and the Pacific Northwest displayed the least segregation. The ten metro/micro areas with the highest indices were scattered around the country in Texas, New York, Iowa, and Virginia. The area with the highest index (0.729) was Pearsall, Texas. The ten areas with the lowest indices were also scattered around the country but leaned towards the West. The lowest index was in Shelton, Washington, at 0.136. It was followed by Winnemucca, Nevada, with 0.167.

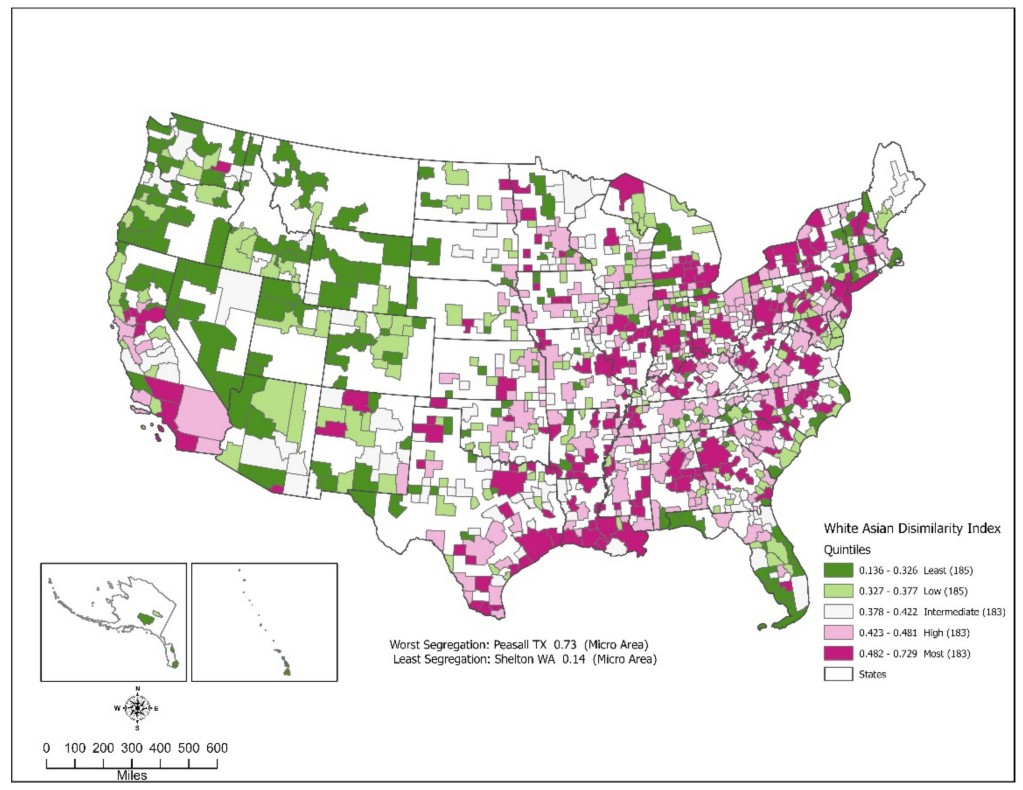

**Figure 16.** Asian-White Dissimilarity Indices, Metro/Micro Areas, 2020.

### 3.2.3. Isolation Index

The Isolation Index, as a measure of the proportion of a racial group living in the same block group, measures the degree to which a member of a racial group is likely to interact with other members of the same group, i.e., intra-group interaction. The alternative of this

is the Interaction Index which measures the probability that a member of a racial group will interact with a member of a different racial group, i.e., inter-group interaction. Massey and Denton elucidate:

> "*Residential exposure refers to the degree of potential contact, or the possibility of interaction, between minority and majority group members within geographic areas of a city. Indices of exposure measure the extent to which minority and majority members physically confront one another by virtue of sharing a common residential area.*" [39]

They then add: "Minority members can be evenly distributed among residential areas of a city, but at the same time experience little exposure to majority members . . . " [39]. A minority, even if residentially integrated, can still be isolated.

For blacks, the median was 0.104, and the mean was 0.178. For all metro and micro areas, blacks had a ten percent chance that they would only interact with other blacks. The range was extensive (0.820), from a low of 0.002 to 0.822. In micro areas, blacks were not as isolated, with an Isolation Index of 0.063, whereas the metro areas had a median of 0.192. The micro areas also displayed the most variability running from the low of 0.002 to the maximum of 0.822.

As with the Dissimilarity Index, the black isolation indices are higher in the East and comparatively low in the West. Eight of the ten metro/micro areas with the highest Isolation Indices were in the South (including Arkansas as being in the South, all ten were in the South). Five were in Alabama (Figure 17). The metro/micro area with the highest Black Isolation Index was Clarksdale, Mississippi, with an index of 0.822. The probability that a black living there will be interacting only with others of the same race was just under 83 percent. The second highest was Greenville, Mississippi, with an index of 0.804, and the third was Cleveland, Mississippi, with an index of 0.788 (Tables 7 and 8).

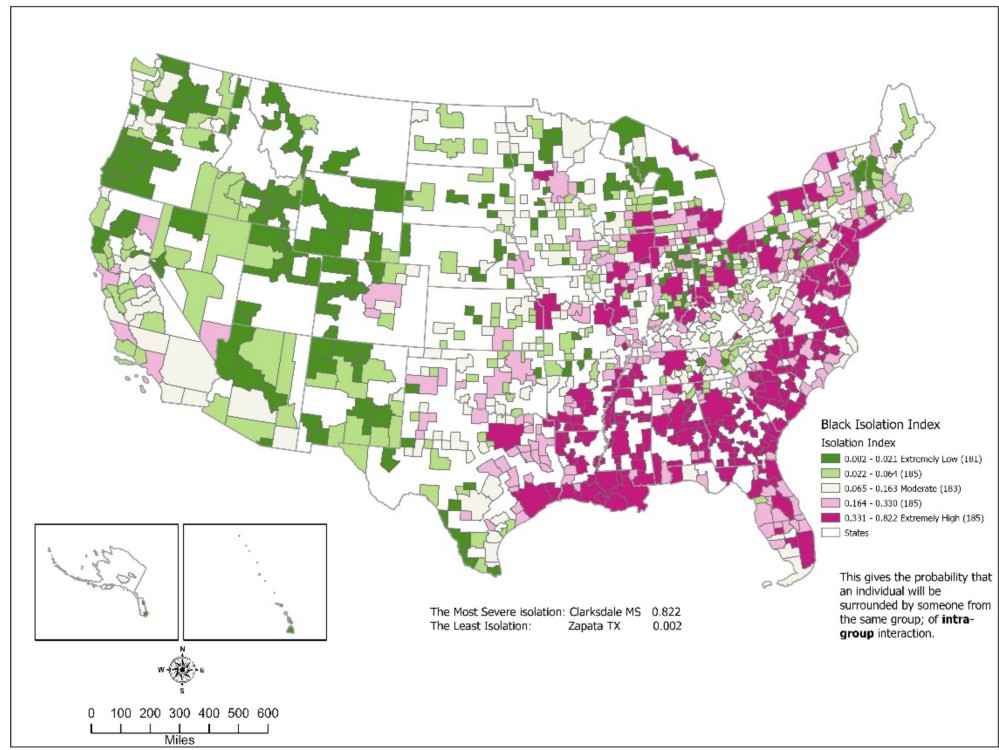

**Figure 17.** Black Isolation Indices, Metro/Micro Areas, 2020.

**Table 7.** Ten Highest Metro/Micro Areas for Black, Hispanic, Native American and Asian Isolation Indices 2020.

| Black Isolation Index (P*) | | Hispanic Isolation Index (P*) | | Native American Isolation Index (P*) | | Asian Isolation Index (P*) | |
|---|---|---|---|---|---|---|---|
| Metro/Micro Area | Index | Metro/Micro Area | Index | Metro/Micro Area | Index | Metro/Micro Area | Index |
| Clarksdale, MS Micro | 0.822 | Rio Grande City, TX Micro | 0.977 | Show Low, AZ Micro | 0.861 | Opelousas, LA Micro | 0.002 |
| Greenville, MS Micro | 0.804 | Laredo, TX Metro | 0.954 | Safford, AZ Micro | 0.86 | Helena, AR Micro | 0.002 |
| Cleveland, MS Micro | 0.788 | Eagle Pass, TX Micro | 0.952 | Gallup, NM Micro | 0.834 | Lafayette, LA Metro | 0.002 |
| Selma, AL Micro | 0.781 | Zapata, TX Micro | 0.939 | Malone, NY Micro | 0.801 | Las Vegas, NM Micro | 0.001 |
| Greenwood, MS Micro | 0.780 | McAllen, TX Metro | 0.929 | Payson, AZ Micro | 0.794 | Burlington, IA Micro | 0.001 |
| Indianola, MS Micro | 0.752 | Brownsville, TX Metro | 0.902 | Grants, NM Micro | 0.732 | Craig, CO Micro | 0.001 |
| Pine Bluff, AR Metro | 0.708 | Nogales, AZ Micro | 0.892 | Alamogordo, NM Micro | 0.711 | Indianola, MS Micro | 0.001 |
| Helena, AR Micro | 0.699 | Raymondville, TX Micro | 0.882 | Española, NM Micro | 0.673 | Pampa, TX Micro | 0.001 |
| Albany, GA Metro | 0.689 | El Centro, CA Metro | 0.873 | Flagstaff, AZ Metro | 0.668 | Lamesa, TX Micro | 0.001 |
| Orangeburg, SC Micro | 0.683 | Pecos, TX Micro | 0.855 | Farmington, NM Metro | 0.663 | Fort Madison, IA Micro | 0.001 |

**Table 8.** Ten Lowest Metro/Micro Areas for Black, Hispanic, Native American, and Asian Isolation Indices 2020.

| Black Isolation Index (P*) | | Hispanic Isolation Index (P*) | | Native American Isolation Index (P*) | | Asian Isolation Index (P*) | |
|---|---|---|---|---|---|---|---|
| Metro/Micro Area | Index | Metro/Micro Area | Index | Metro/Micro Area | Index | Metro/Micro Area | Index |
| Zapata, TX Micro | 0.002 | Mount Gay, WV Micro | 0.009 | Rio Grande City, TX Micro | 0.001 | Raymondville, TX Micro | 0.0003 |
| Rio Grande City, TX Micro | 0.002 | Point Pleasant, WV Micro | 0.01 | Spirit Lake, IA Micro | 0.002 | Orlando, FL Metro | 0.0004 |
| Othello, WA Micro | 0.003 | Selma, AL Micro | 0.011 | Zapata, TX Micro | 0.002 | Boone, NC Micro | 0.0004 |
| Hailey, ID Micro | 0.003 | St. Marys, PA Micro | 0.012 | Carroll, IA Micro | 0.002 | Hinesville, GA Metro | 0.0004 |
| Burley, ID Micro | 0.004 | Jackson, OH Micro | 0.013 | Easton, MD Micro | 0.002 | Daphne-Fairhope-Foley, AL Metro | 0.0004 |
| Vernal, UT Micro | 0.004 | West Point, MS Micro | 0.013 | Johnstown, PA Metro | 0.002 | San Antonio, TX Metro | 0.0004 |
| Hood River, OR Micro | 0.004 | Warren, PA Micro | 0.013 | St. Marys, PA Micro | 0.002 | Raleigh-Cary, NC Metro | 0.0005 |
| Mountain Home, AR Micro | 0.004 | Brookhaven, MS Micro | 0.013 | Grenada, MS Micro | 0.002 | Pearsall, TX Micro | 0.0005 |
| Blackfoot, ID Micro | 0.004 | Elkins, WV Micro | 0.015 | Somerset, PA Micro | 0.002 | Lakeland, FL Metro | 0.0005 |
| Coeur d'Alene, ID Metro | 0.004 | Coshocton, OH Micro | 0.015 | Lebanon, PA Metro | 0.002 | Fayetteville, AR Metro | 0.0005 |

Hispanics in metro/micro areas had a median Isolation Index of 0.114 and a mean of 0.182. Substantiating generally low isolation of Hispanics, twenty-five percent of the areas had an index of 0.052 or lower, and twenty-five percent had an index of 0.236 or greater. The Hispanic indices had the greatest range (0.968), running from a low of 0.009 up to 0.977, the highest Isolation Index of the 927 metro/micro areas. Hispanics in Metro areas tended to be more isolated (median index of 0.150) than Hispanics in Micro Areas (median index of 0.084).

The geographic distribution of the indices indicates that the most extreme Hispanic Isolation Indices (an index equal to or greater than 0.295) were in the Southwest, Southern California, Florida, and the northern East Coast (Figure 18). Eight of the ten metro/micro areas' highest Isolation Indices were in Texas, one in Arizona, and one in California. For Texas, most of the areas having a high index are on the Mexican border, as is the area in Arizona and California. The area with the highest Hispanic Isolation Index was Rio Grande City, Texas, with an index of 0.977. Laredo, Texas, was the second highest with 0.954. The Rio Grande, Texas, index means that Hispanics living there were, on average, residing in block groups that were 98 percent Hispanic; they have a 98 percent chance of only interacting with other Hispanics. The areas with the lowest Hispanic Isolation Indices are scattered around the county. The lowest Index was found in Mount Gay-Shamrock, West Virginia, with an index of 0.009; the next lowest was also in West Virginia (Point Pleasant), with an index of 0.010 (Tables 7 and 8).

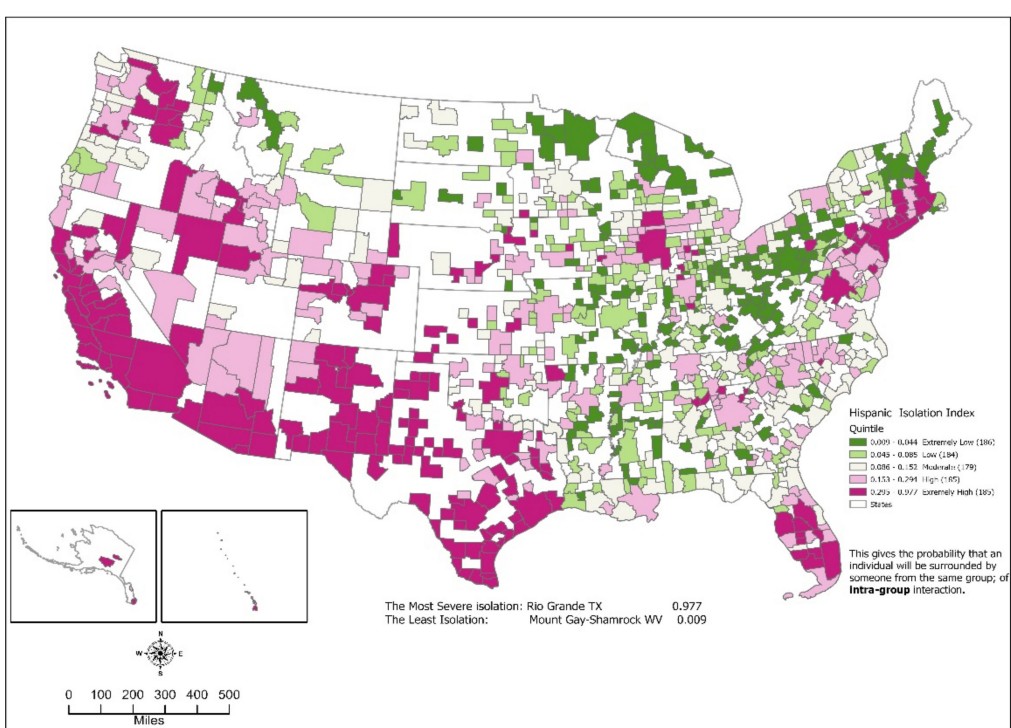

**Figure 18.** Hispanic Isolation Indices, Metro/Micro Areas, 2020.

Native American Isolation Indices ranged from a low of 0.001 to a maximum of 0.861. The median index was 0.006, and the mean was 0.038. Metro areas had a median index of 0.388. The high was 0.773. Native Americans living in micro areas were less isolated, with a median index of 0.005, though the maximum micro area was 0.861. Overall, Native American Isolation Indices were low, with twenty-five percent having an index of 0.004 or less and twenty-five percent having an index of only 0.012 or greater.

Geographically, the higher Native American Isolation Indices were clustered in the Southwest, the upper Plains, and the West (Figure 19). The highest index was for Show Low, Arizona, followed by Safford, Arizona, with indices of 0.861 and 0.860, respectively. Eight of the areas with high indices were in Arizona and New Mexico. Though four were in Pennsylvania, areas with low indices were scattered around the county. The area with the lowest index was Rio Grande, Texas, at 0.001, followed by Spirt Lake, Iowa, at 0.002 (Tables 7 and 8). Oklahoma again shows up as an anomaly. Most of the state's metro/micro areas are found in the highest twenty percent. This is along with the exceptionally low Native American dissimilarity indices. Keep in mind that the isolation indices are also extremely low in an absolute sense. The situation in Oklahoma, where Native Americans had low residential segregation but were isolated, is clearly visualized. In Arizona and New Mexico, The Native American is highly segregated and isolated.

Asian Isolation was extremely low, with a median index of 0.001 and a mean index of 0.001. The lowest index was 0.0004, but the highest index was 0.002. Twenty-five percent of the areas had an index of 0.0007 or less, while twenty-five percent had an index of 0.0008 or greater. Metro and micro areas displayed the same levels of isolation.

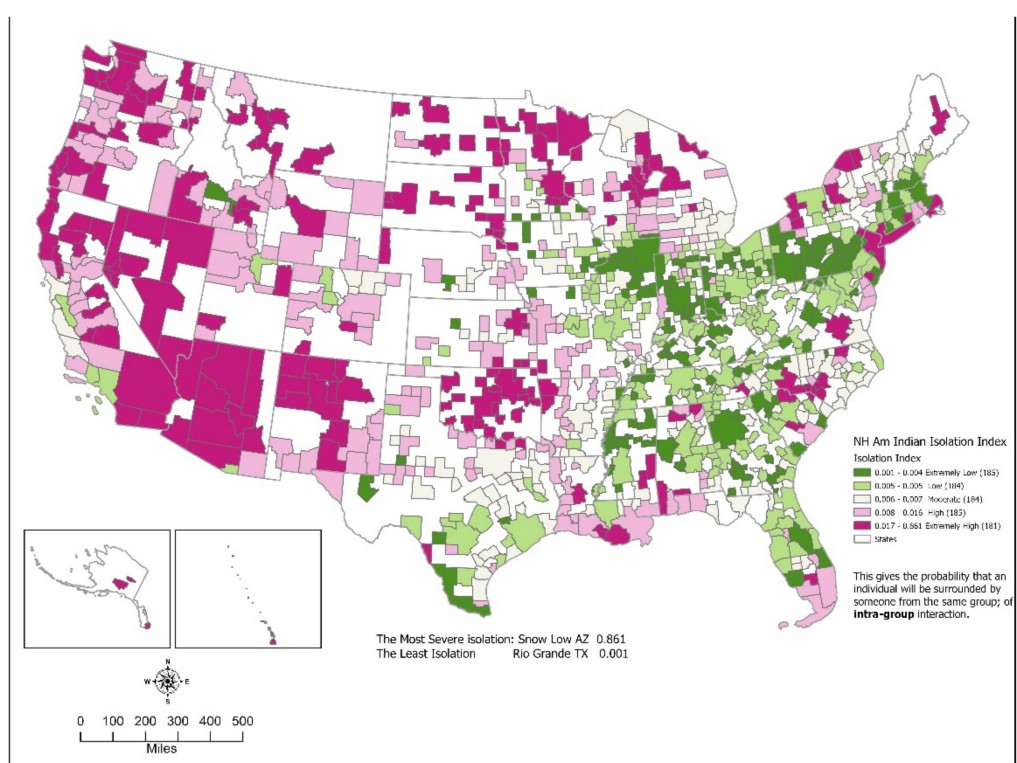

**Figure 19.** Native American Isolation Indices, Metro/Micro Areas, 2020.

The geographic distribution of Asian Isolation Indices displayed some variation (Figure 20). Isolation was highest in the Great Lakes region, the Rustbelt and the northern Midwest, and the pacific northwest. California had several areas with high isolation. The South and most of the Southwest displayed the least Asian isolation. The ten metro/micro areas with the highest indices were scattered around the country in Texas, Arkansas, Iowa, and Louisiana. The area with the highest index (0.002) was Opelousas, Louisiana. The ten areas with the lowest indices were also scattered around the country but leaned towards the South. The lowest index was in Raymondville, Texas, at 0.0003. It was followed by Orlando, Florida, with 0.0004 (Tables 7 and 8).

Note that all the distributions of the isolation Indices for all racial groups are either highly skewed (usually to the right), high kurtosis (usually very peaked), or both. They are not symmetrical. One of the practical effects of this is to make comparing the racial groups complex: what is extreme for one racial group is not for another.

### 3.3. Relationship between Racial Diversity & Racial Segregation

The relationship between racial diversity and racial residential segregation was found to be very complex. Looking at Diversity Indices (D) and Multigroup Dissimilarity Indices ($D_G$), a positive, linear relationship was found with a correlation coefficient of 0.422 with $p = 0.000$. The slope was +0.249. However, the question of statistical significance is moot as this is applied to the entire population of metro/micro areas (Figure 21).

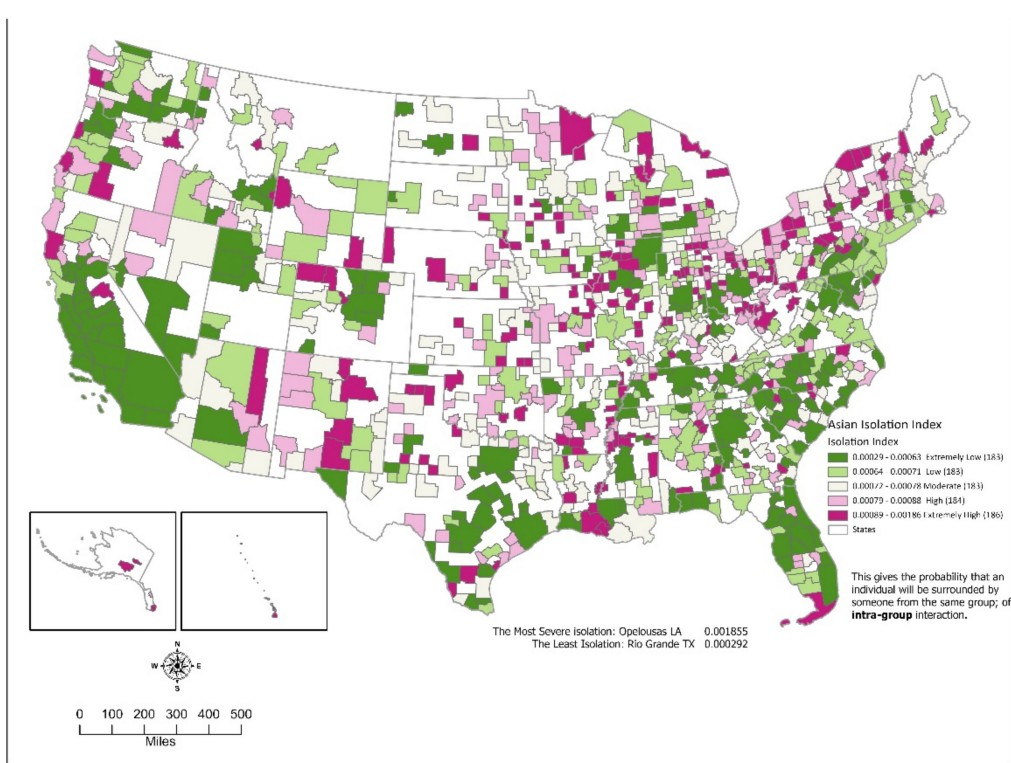

**Figure 20.** Asian Isolation Indices, Metro/Micro areas, 2020.

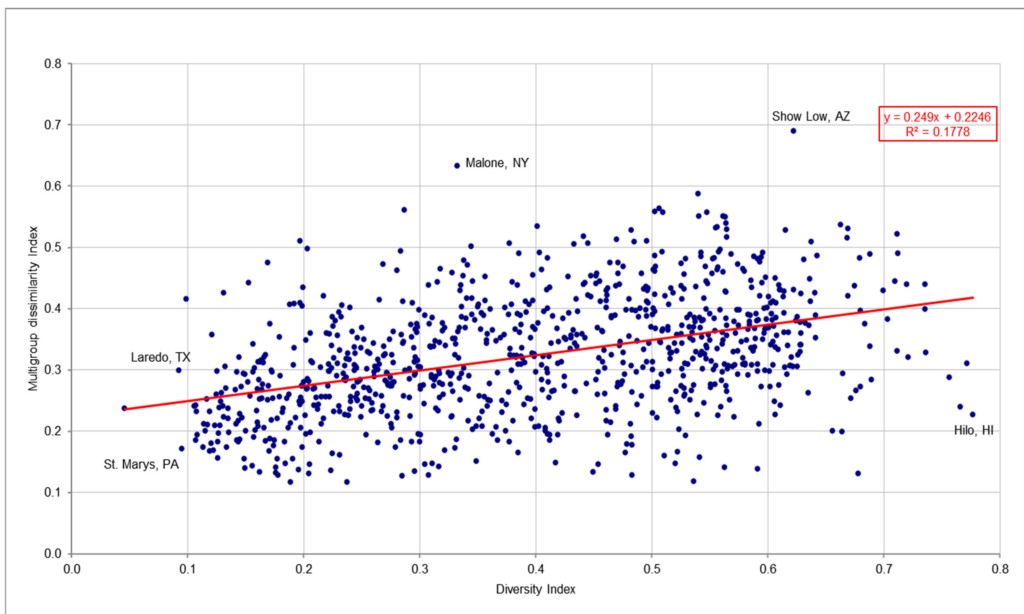

**Figure 21.** Relationship Between Diversity and Multigroup Dissimilarity.

The relationship between diversity and pair-wise dissimilarity varied by racial group. For Black-white dissimilarity, the relationship was very weak. The correlation coefficient was only 0.084 and had a slight positive upward slope (+0.058) (Figure 22). In contrast, the relationship between diversity and Hispanic-White dissimilarity was strong, with a correlation coefficient of 0.364 and a positive slope of +0.225 (Figure 23).

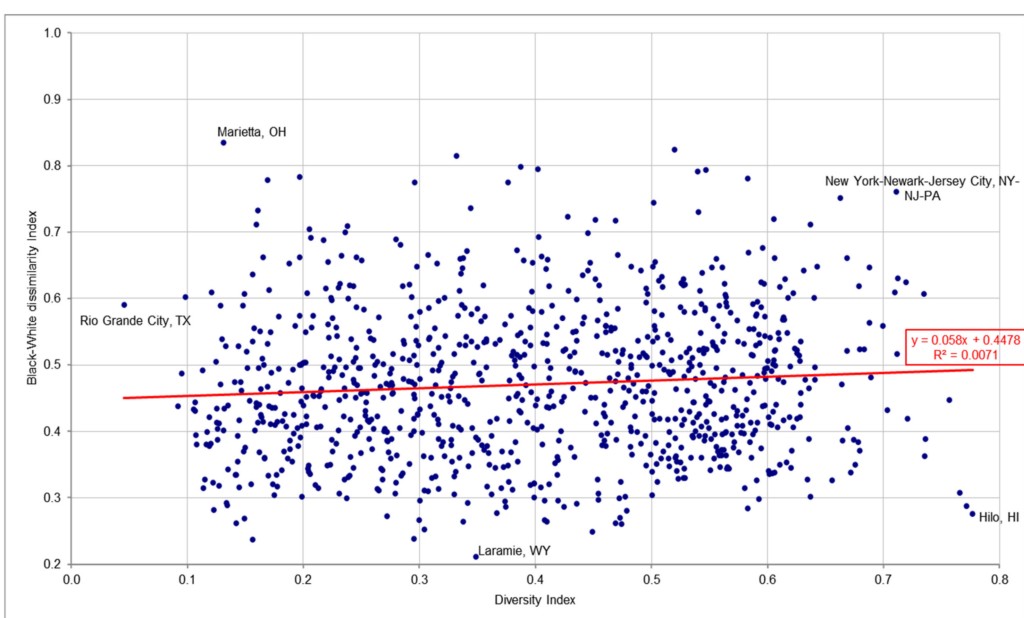

**Figure 22.** Relationship Between Diversity and Black-white Dissimilarity.

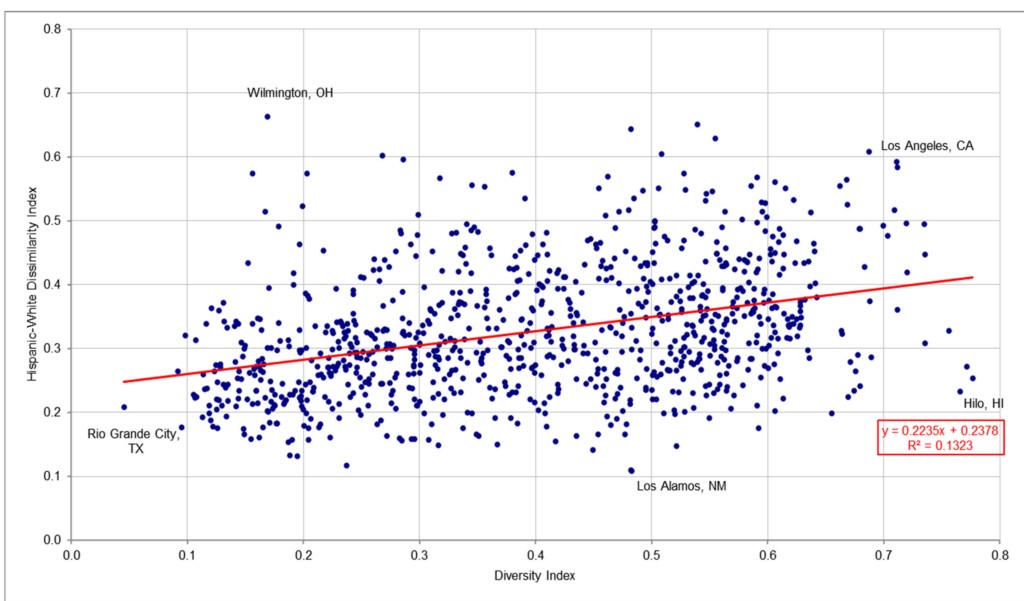

**Figure 23.** Relationship Between Diversity and Hispanic-White dissimilarity.

Native American-White dissimilarity and diversity had a weak and inverse relationship. The correlation coefficient was −0.052. The slope was −0.00384. Thus, the higher the diversity index, the lower the Native American-White Dissimilarity Index (Figure 24).

Asian-white Dissimilarity and diversity had a correlation coefficient of 0.127 and a slope of 0.0706. The relationship was positive but weak (Figure 25).

The relationship between racial diversity and racial isolation was more consistent. Black isolation and diversity had a correlation coefficient of 0.482. The relationship had a solid positive slope of 0.554. The greater the diversity of metro/micro areas, the more isolated blacks were (Figure 26).

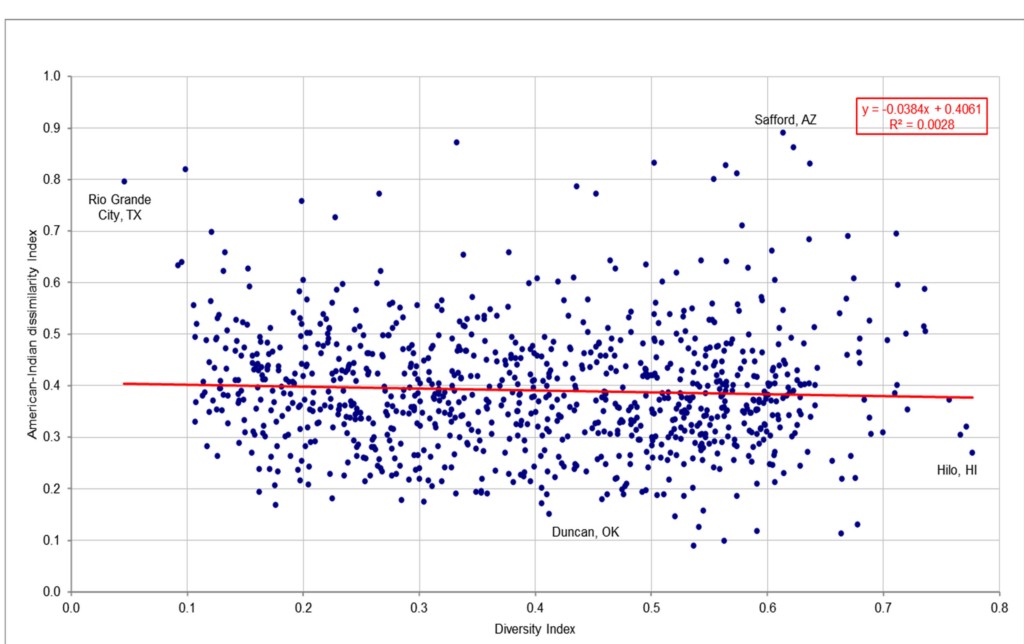

**Figure 24.** Relationship Between Diversity and Native American-White Dissimilarity.

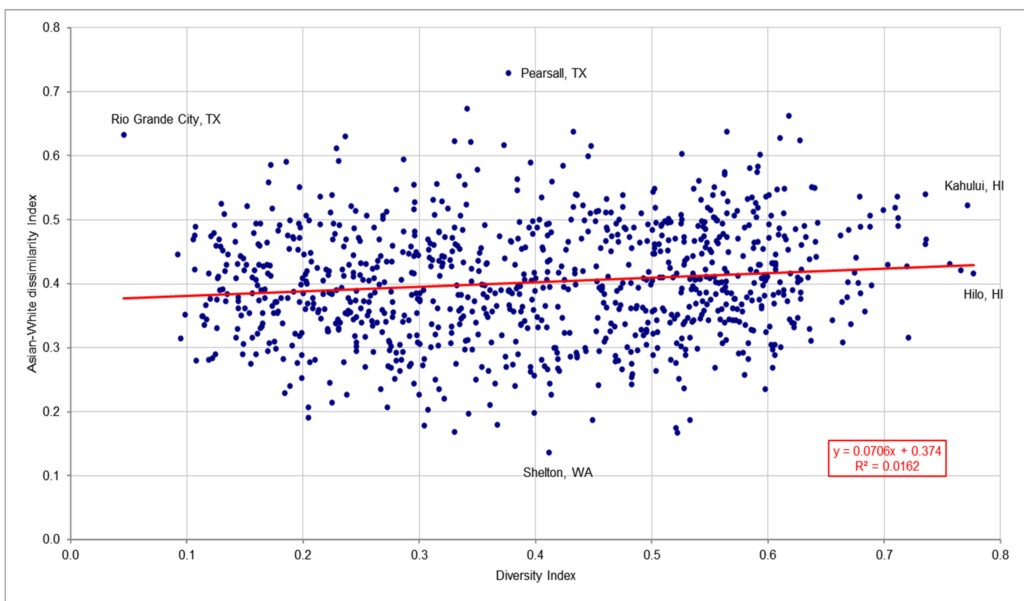

**Figure 25.** Relationship Between Diversity and Asian-White Dissimilarity.

Hispanic isolation was also strongly related to diversity, with a correlation coefficient of 0.413. The slope was a positive 0.478. The relationship produces a bowed curve in the distribution due to some metro/micro areas having very low racial diversity but very high Hispanic Isolation Indices. This situation is generated by the areas having very few whites. This upper strip in Figure 27 are areas such as Laredo, TX, and Nogales, AZ, along the Mexican border.

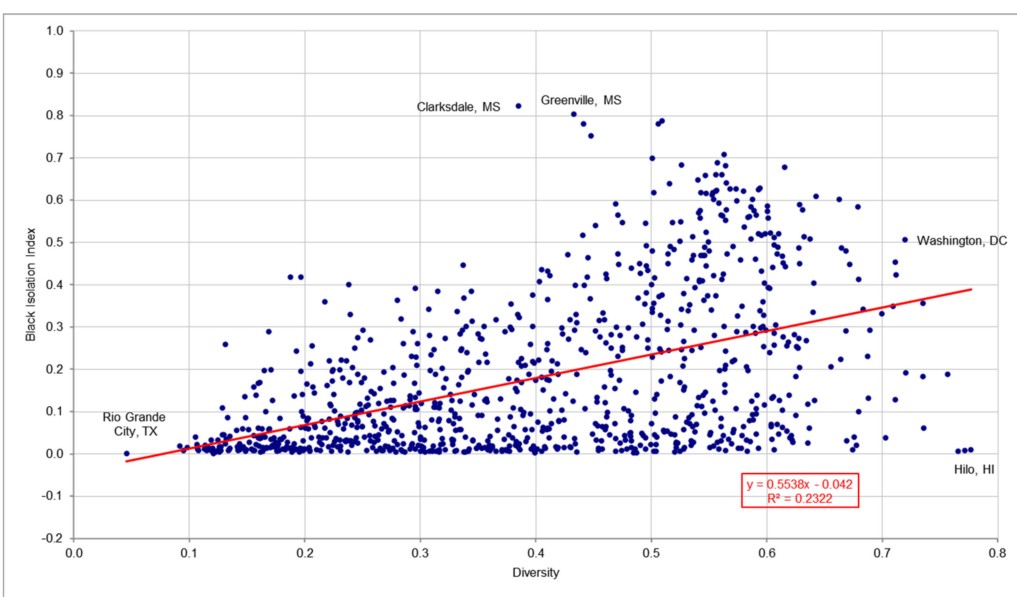

**Figure 26.** Relationship Between Diversity and black Isolation.

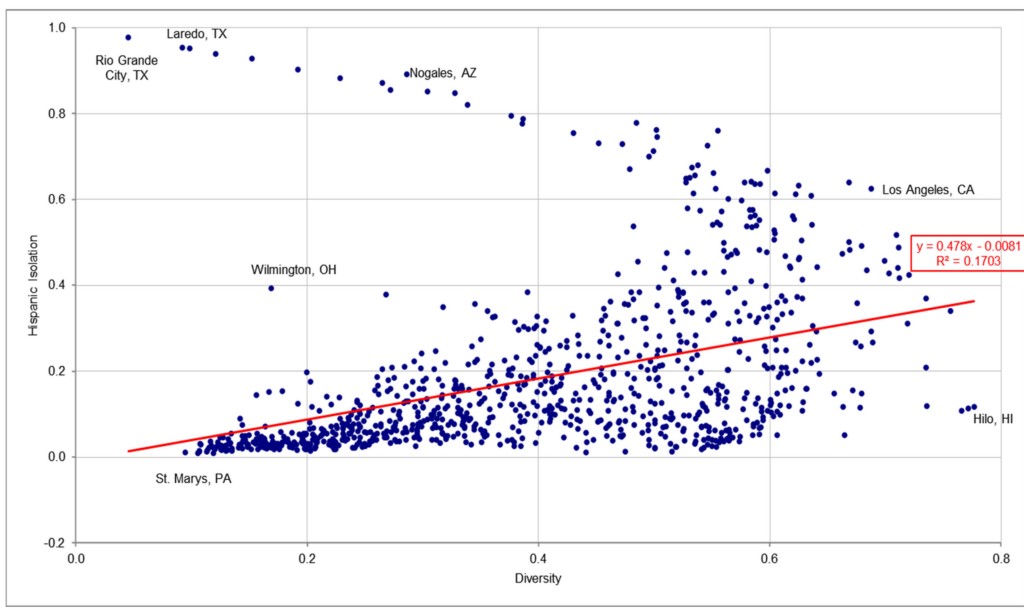

**Figure 27.** Relationship Between Diversity and Hispanic Isolation.

The relationship between diversity and Native American isolation was relatively weak, with a correlation coefficient of 0.155 and a positive slope of 0.1055. Overall, the isolation displayed is due to the historically forced segregation and isolation of the Native Americans into reservations or remaining in historical territories (Figure 28).

The relationship between diversity and Asian isolation was strong, but it was inverse. The correlation coefficient was −0.285, and the slope was −0.0003. The greater the racial diversity index, the Asians were less isolated. No other racial group had an inverse relationship between diversity and isolation (Figure 29).

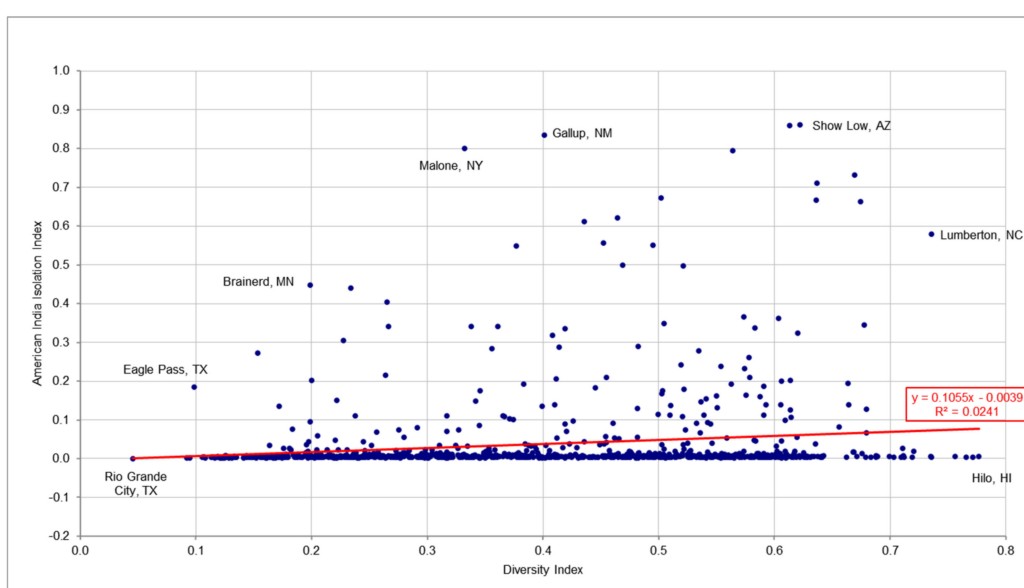

**Figure 28.** Relationship Between Diversity and Native American Isolation.

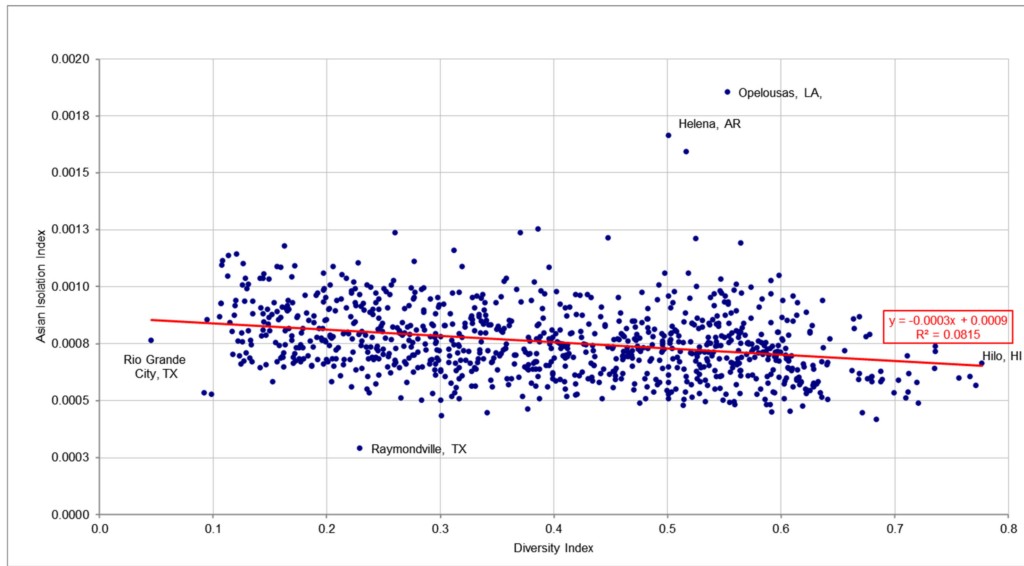

**Figure 29.** Relationship Between Diversity and Asian Isolation.

## 4. Conclusions

The two facets of racial residential segregation examined within metropolitan and micropolitan areas at the block group level (spatial dispersion and isolation) were related to the metropolitan and micropolitan area's racial diversity though the relationship is complex.

1.  The racial diversity indices had a bimodal distribution. The median index was 0.399 with an interquartile range of 0.276 and a broad range of 0.731. The South was the most diverse, while the northeast and Great Lakes region were the most homogenous.
2.  The Multigroup Dissimilarity Index had a median index of 0.318, an interquartile range of 0.131, and a broad range of 0.573. Multigroup dissimilarity was unevenly distributed geographically focused on the eastern half. Five areas had extreme values, over 2.5 standard deviations from the mean. The overall distribution of the races within the 927 metropolitan and micropolitan areas was uneven (i.e., "lumpy").
3.  The differences in the pairwise racial dissimilarity indices clearly show the "lumpiness" of segregation.

a.  The Black-white dissimilarity index was the highest, thus displaying more residential segregation. The micropolitan areas were slightly less segregated than the metropolitan areas (medians were 0.437 and 0.499, respectively).

b.  Hispanic-white dissimilarity was relatively low, with a median of 0.312. The Hispanic dissimilarity ran from a low of 0.108 to a high of 0.663. The micropolitan areas (median = 0.292) were significantly less segregated than the metropolitan areas (median = 0.352). The highest indices were found in the country's eastern half though California and New Mexico were also in the highest group.

c.  The Native American-white dissimilarity index was moderate, with a median of 0.380. Indices ran from a low of 0.091 to a high of 0.891. A relatively large number of areas (nine) fell in the extreme range. Geographically, the Native American indices were highest East of the Mississippi River and in the far Southwest. The indices were basically the same for metropolitan areas (median = 0.388) and the micropolitan areas (median = 0. 373). Oklahoma was an anomaly, with nine of the ten least segregated areas.

d.  Asian-white dissimilarity was high. The median was 0.399, with values ranging from a low of 0.136 to a high of 0.729. Segregation was highest in the Mid-Atlantic/Northeast, Midwest, and the South. California had several areas with high segregation. The Mountain region and the Pacific Northwest displayed the least segregation.

4.  The Isolation Index indicating the probability that a person chosen at random will interact with other group members in their block group had low indices in an absolute sense. There was a great deal of variation among each group and between the racial groups.

a.  The black isolation index had a median of 0.104, varying from a low of 0.002 to a high of 0.822. Areas with high indices were concentrated in the eastern half of the country. The metropolitan areas had higher indices than the micropolitan areas (medians of 0.192 and 0.063, respectively). All ten areas with the highest indices were in the South, and five were in Mississippi. The lowest indices were found in the western half of the country. There was a substantial overlap between Black-white residential segregation and black isolation. So, blacks were not only clumped together across the areas but were also isolated into the same block groups they would thus have more interaction and contact with other blacks.

b.  The Hispanic isolation indices were slightly higher with a median of 0.114, ranging between 0.009 and 0.997. This was the largest range of values for any racial group. Metropolitan areas had higher indices (median = 0.150) than micropolitan areas (median = 0.084). The higher Hispanic isolation indices were geographically concentrated on the west coast and the Southwest, though Florida and the northeast coastal areas registered high values. Eight of the areas with the highest isolation indices were in Texas. All ten were found along the Mexican border. The west coast, particularly California, displayed the most areas with a substantial overlap of white-Hispanic dissimilarity and Hispanic isolation.

c.  Native American isolation indices were among the lowest. The median index was 0.006, ranging from a low of 0.001 to a high of 0.861. Metropolitan and micropolitan areas had basically the same median index, 0.006 and 0.005, respectively. Geographically, the highest Native American isolation indices were found in the West but were also clustered in the north-central states along the Canadian border, Oklahoma, North Carolina, and New York. These areas are the locations of reservations and traditional and historical tribal regions. Nine of the ten highest areas were in Arizona and New Mexico, with the other area being in New York. Oklahoma again was an anomaly, with relatively high Native American isolation indices while having relatively low dissimilarity indices.

> Several metro/micro areas in Arizona and New Mexico had significant overlap between Native American-white dissimilarity indices and the isolation indices. In these areas, Native Americans were not only clumped together, but they also had a high probability of only interacting with other Native Americans.
>
> d.   Asian isolation was exceptionally low. The median was 0.001. Isolation was highest in the Great-lake region, the Rustbelt and the northern Midwest, and the pacific northwest. California had several areas with high isolation. The South and most of the Southwest had displayed the least Asian isolation.

5.   The racial diversity index was significantly related to the multigroup dissimilarity index as hypothesized (H1). The correlation coefficient was 0.422 with a slope of 0.249. As racial diversity increases within the metro/micro areas, so does the overall racial residential segregation amongst the eight racial groups.

6.   The relationship between racial diversity and the pairwise dissimilarity indices varied by racial group.

   a.   The relationship between the Black-white dissimilarity index and racial diversity was linear but weak, with a correlation coefficient of 0.084 and a slope of 0.058. The hypothesis was not supported.

   b.   Racial diversity and the Hispanic-white dissimilarity index were significantly related. The hypothesis (H2) of the relationship was strongly supported for this racial group. The correlation coefficient was 0.364, and the slope was 0.224. The connection was linear.

   c.   Hypothesis (H2) is not supported other than very, very weakly. The Native American-white dissimilarity index was not significantly related to racial diversity. The correlation coefficient was −0.052, and the slope was −0.038. The relationship was inverse: the higher the racial diversity, the lower the racial residential segregation.

   d.   Asian-white dissimilarity was significantly related to racial diversity. The correlation coefficient was 0.127 with a slope of 0.706. Hypothesis (H2) is supported for this racial group.

7.   The relationship between racial diversity and racial isolation also varied by racial group, though very strong positive correlations were found. Hypothesis H3 was supported.

   a.   In contrast to diversity and the Black-white dissimilarity index, urban racial diversity and black isolation were significantly related. The relationship was linear, and the correlation coefficient was 0.482 with a slope of 0.554. Hypothesis H3 was confirmed for this group. The greater the racial diversity of the metro/micro areas, the greater the isolation of blacks.

   b.   Racial diversity was also found to be significantly related to Hispanic isolation. The relationship was linear, with a correlation coefficient of 0.413 and a slope of 0.478. for Hispanics, hypothesis H3 was strongly supported. Here, too, the greater the urban racial diversity, the greater the isolation of Hispanics.

   c.   For Native Americans, urban racial diversity was not significantly related to Native American isolation. What relationship existed was weak, so H3 was only weakly supported. It was linear, the correlation coefficient was 0.155, and the slope was 0.1055.

   d.   For Asians, racial diversity was strongly related to Asian isolation (correlation coefficient of −0.285 and a slope of −0.00003). The relationship was the only inverse relationship between diversity and isolation—the greater the racial diversity, the lower the isolation of Asians.

A connection was found between racial diversity of metro/micro areas and racial segregation and racial isolation at the block group level. This connection has been lacking in the research. The most general conclusion is that there was a stronger and more consistent relationship between racial diversity and racial isolation than between residential racial dissimilarity and racial diversity. The Multigroup racial Dissimilarity Index was strongly

related to racial diversity: the higher the metro/micro racial diversity, the higher the multi-group segregation. Racial isolation was also strongly related to racial diversity: The higher the metro/micro racial diversity, the greater the racial isolation. The exception was Asians, where the connection was inverse: the higher the racial diversity, the lower the isolation of Asians. For pairwise racial dissimilarity, the relationship to racial diversity was supported but most strongly for Hispanics: The higher the racial diversity, the greater the segregation of Hispanics from whites. Table 9 displays the statistical relationships found. Appendix A presents four maps which display the geographic overlap of these relationships.

**Table 9.** The Hypothesis and Related Conclusions.

| | HYPOTHESIS | Group | Correlation | Slope | Relationship | |
|---|---|---|---|---|---|---|
| $H_1$ | Within Metropolitan and Micropolitan Areas and at the small grain level of block groups, the greater the degree of population differentiation on the characteristic of race as measured by the diversity Index (DI), the greater the level of residential segregation as measured by the Multigroup Index of Dissimilarity (Dg). | all 8 | r = 0.422 | 0.249 | linear | strongly supported |
| $H_2$ | Within Metropolitan and Micropolitan Areas and at the small grain level of block groups, the greater the degree of population differentiation on the characteristic of race as measured by the diversity Index (DI), the greater the level of residential segregation as measured by the Index of Dissimilarity (D). | Black | r = 0.084 | 0.058 | linear | weak |
| | | Hispanic | r = 0.364 | 0.224 | linear | strongly supported |
| | | Indian | r = −0.052 | −0.038 | inverse | weak |
| | | Asian | r = 0.127 | 0.706 | linear | supported |
| | | Pacific | r = −0.221 | −0.280 | inverse | supported |
| | | Other | r = −0.116 | −0.059 | inverse | weak |
| | | Mixed | r = 0.213 | 0.071 | linear | supported |
| $H_3$ | Within Metropolitan and Micropolitan Areas and at the small grain level of block groups, the greater the degree of population differentiation on the characteristic of race as measured by the diversity Index (DI), the greater the level of residential isolation as measured by the Isolation Index (P*). | Black | r = 0.482 | 0.554 | linear | strongly supported |
| | | Hispanic | r = 0.413 | 0.478 | linear | strongly supported |
| | | Indian | r = 0.155 | 0.106 | linear | weak |
| | | Asian | r = −0.285 | −0.0003 | inverse | supported |
| | | Pacific | r = 0.193 | 0.018 | linear | supported |
| | | Other | r = 0.149 | 0.007 | linear | weak |
| | | Mixed | r = 0.184 | 0.024 | linear | supported |

Two topics for further research are: (1) Is it possible to estimate or measure the point between voluntary segregation and involuntary segregation on a macro scale? (2) Explore and explain the causes of the geographic distribution of the segregation indices (multigroup and each pairwise indices) and the geographic distribution of the racial isolation indices.

**Funding:** This research received no external funding.

**Acknowledgments:** The author would like to acknowledge the editing and assistance of Mary Claire Cooperrider.

**Conflicts of Interest:** The author declares no conflict of interest.

## Appendix A

The following maps visualize the congruence of racial diversity, racial segregation, and racial isolation. The colors are blended to produce new unique shades of color. The maps display the Metro/Micro areas based upon the values of the respective indices. The higher the values are, the darker the colors. The following maps (Figures A1–A4) were based upon a quartile classification of each variable.

Figure A1 visualizes the joint distribution of racial diversity and multigroup racial segregation. The lightest blue-purple shows those areas where both racial diversity and multigroup racial diversity indices were low. The darkest wine color shows where BOTH indices were high: racial diversity AND multigroup racial segregation were high. Visibility, these metro/micro areas fall along the East Coast, the Gulf Coast, and California. Additionally, Chicago, Detroit, and Milwaukee join the list.

Figure A2 displays the joint distribution for Metro/Micro areas on Black-white segregation and black isolation indices. On this map, the darkest blue show where BOTH indices are high: high black-white segregation AND high black isolation. This suggests that blacks are both segregated from whites and are isolated. The South and the east coast stand out in this regard. Additionally, several major Metro areas show up here: Chicago, Detroit, Milwaukee, St Louis, and Cleveland.

Figure A3 displays the joint distribution for Metro/Micro areas on white-Native American segregation and Native American isolation indices. These are the areas where native Americans were forced into reservations. The anomaly of Oklahoma stands out as the state has low White-native American segregation but relatively high native American isolation. The Southwest was the prominent region where Native Americans were both segregated and isolated.

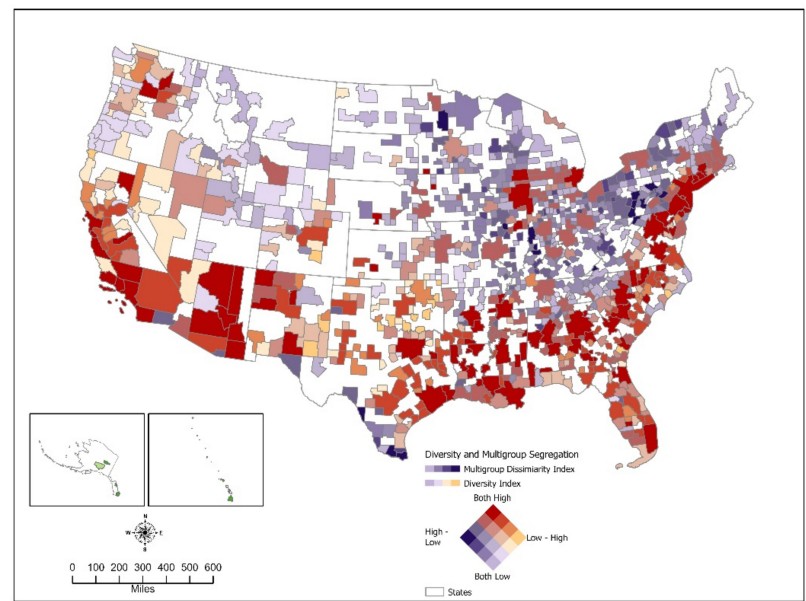

**Figure A1.** Joint Distribution of Racial Diversity and Multigroup Dissimilarity, 2020.

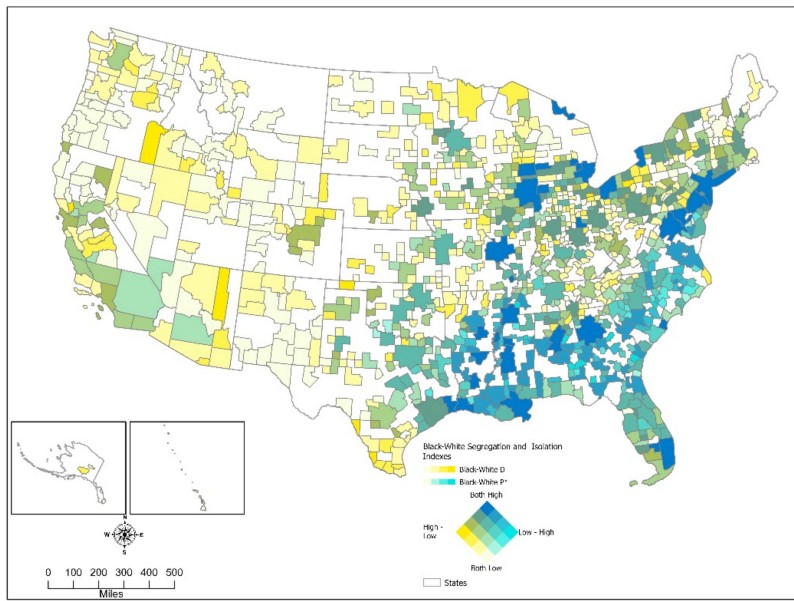

**Figure A2.** Joint Distribution of Black-White Segregation and Black Isolation, 2020.

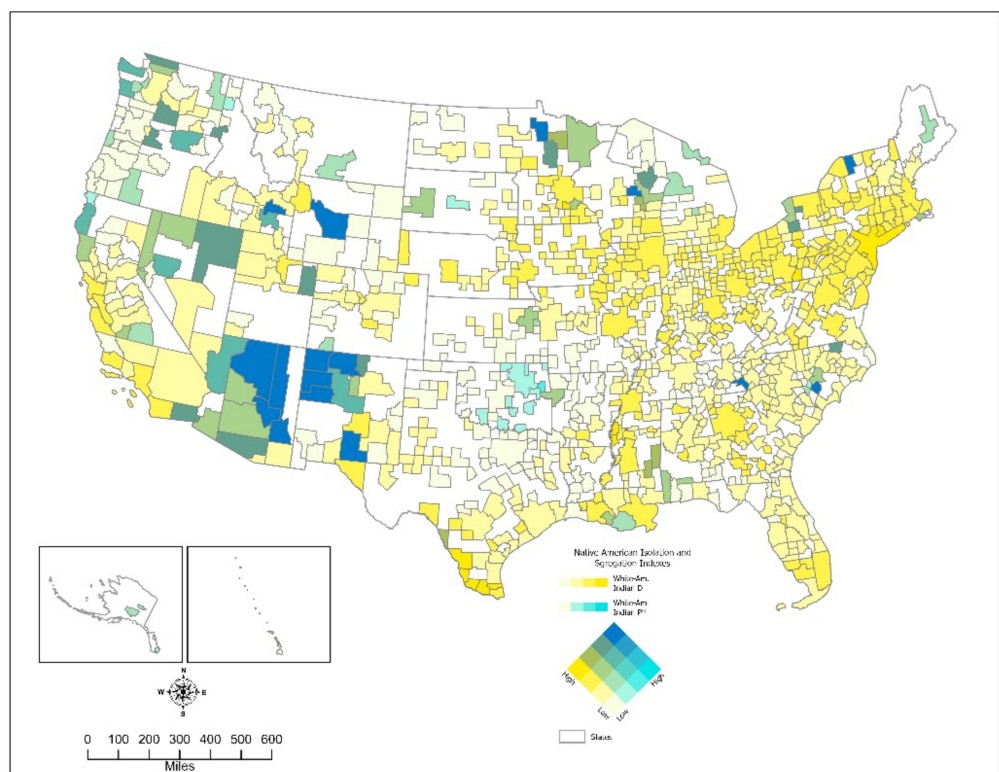

**Figure A3.** Joint Distribution of White-Native American Segregation and Isolation, 2020.

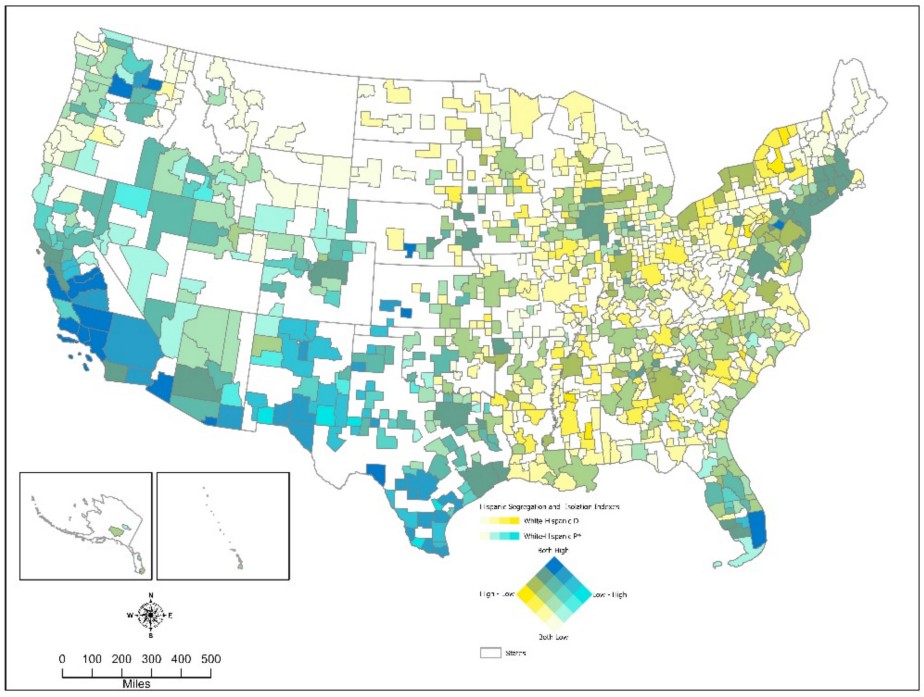

**Figure A4.** Joint distribution of White-Hispanic Segregation and Hispanic Isolation, 2020.

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
