# Peer review of "The Relationship between Urban Diversity and Residential Segregation"

_urbansci, doi:10.3390/urbansci6040066_

Round 1

Reviewer 1 Report

This is a brilliant article, covering an enormous range of relevant material with the most detail I have ever seen in a comparable study. The conclusions are nuanced and fascinating.

Author Response

Reviewer #1

Thank you for your time in reviewing my manuscript. I appreciate your comments. I did try to provide details but not so much that it veered off subject, which I'm sure you are aware can happen easily (sometimes called "mission-creep").   

Again, thank you.

Reviewer 2 Report

I think that this paper is interesting and should be considered for publication. The strongest parts of the paper are its methods, the data interpretation as well as the maps drawn in GIS. In my opinion there are three issues which need to be revised or expanded in the paper:

1) Theoretical section has to be enlarged and updated a bit, mainly for aspects connected to urban segregation and diversity.
First, the author should link their article to recent works on urban segregation in the USA, Europe and in other worldwide urban areas in order to better position the study in the international literature of diversity and segregation. A recent study of Buck Kyle et al, 2021 on investigating the relationship between environmental qualitysocio-spatial segregation and the social dimension of sustainability in US urban areas was published in journal Sustainable Cities and Society. For Europe there were case studies on the segregated Roma people which can be mentioned too, see examples from Hungary and other countries - Gyorgy Malovics et al (2020) in Geographica Pannonica who presented issues of desegregation in relation to the perpetuation of stigma for some segregated Roma communities as well as Boglarka Mereine Berki et al (2017) in Journal of Urban and Regional Analysis who wrote about social capital and poverty alleviation issues in segregated urban contexts. Furthermore, Berki et al. (2021) in journal Cities presented a connection between segregated people, social mixing, social capital and brought a new dimension in segregated urban studies. For the Czech Republic and Romania there has been a study in journal IJURR (2022) on urban encounters and fragmented habitus for the poor segregated Roma people as well.

As segregation usually leads to territorial stigmatization, it can be mentioned that sustainable urban development could be hardly implemented if such territorial stigma appears in urban outcasts and in urban segregated areas - several studies should be given as examples in this sense, see for instance Alistair Sisson's work in Progress in Human Geography (2020) or other papers published in International Journal of Urban and Regional Research in 2018 (e.g the study of Ryan Powell on segregated wealthy Roma people) and in 2019 (highlighting far right populism against the segregated urban people. In relation to these aspects, Loic Waquant's works on urban outcasts and segregation could be shortly mentioned too. On the other hand, there were cases when segregated poor people are received as the others or as otherness in multiethnic neighbourhoods (see Covaci and Jucu, 2021 in journal Identities).

In conclusion, there is a need to expand paper's literature review with examples as the ones presented above and even several more dealing with diversity and segregation.

2) There is no discussion section in this paper. It would be recommended to draw in this paper some discussions (which have to connect paper's findings with the literature review) and even some policy recommendations. 

3) Conclusions should say more about the implications of this study (what it brought new to what we already know - so some international works have to be mentioned and how this paper pushed further what has been published by now) and some follow-up research (or how other scholars can further develop the outcomes of this paper).

Reviewer 3 Report

Overall comments: This paper aims to explore how the diversity in urbanized areas impacts residential segregation in those areas. The author uses several segregation indices in their analyses and includes all metro and micropolitan areas. They find a relationship between diversity and forms of segregation. These findings have the potential to be important for segregation researchers. However, I have a series of suggestions that would strengthen the paper and add to the conclusions the author is able to draw.

Suggestions:

The introduction could be expanded upon, it feels so brief. Perhaps an overview of the subsections in the rest of the introduction would orient readers better. The introduction currently reads choppy and disjointed. Transitions would help guide the reader through the sections and make clear why they are included.

While the discussion of the definition of urban is likely useful for those familiar with urban studies and Census definitions, I think it could be expanded upon and clarified for those less familiar with these conceptualizations. Census conceptualizations are confusing, and some readers may need more guidance here. Also, it is not completely clear how this relates to the author’s paper. I would encourage the author to streamline to the parts most relevant to these analyses.

While the introduction has some good information, it does not currently clearly lay out the purpose of this manuscript or illustrate how it fits with previous literature. Reframing this to more clearly lay out what we know about this topic and how this paper fits in will help readers understand the contributions from the author.

I’m not clear which data the author is using. Are they ACS data? Decennial Census data? It’s never stated. The author does a nice job of explaining how they accessed the data, but it’s not clear what data are being used. I think adding a Data section at the beginning of the methods section would help with this.

The way the paper is written, it assumes diversity is desired by and good for all groups of people.  I’m certainly not saying it isn’t, but the author needs to justify/interrogate this idea more. I would suggest reviewing the residential preferences literature (see work by Maria Krysan to start). Also, this paper doesn’t consider variation in outcomes by racial/ethnic composition of city and/or block groups. I would also encourage the author to think about how racism impacts any of these outcomes. This is alluded to in the discussion of Indigenous segregation, but not explicitly explored.

The author loosely touches on the idea of how racial/ethnic composition of an area and spatial dispersion of racial/ethnic groups across the country might impact their findings but does not really discuss this. Also, I would like to see a discussion of why the findings vary by race/ethnicity, what the importance of the findings are, how this adds to our understanding of residential segregation. The author does not include any control variables in their analyses despite there being a large body of literature identifying how other sociodemographic characteristics are related to segregation. Even if the author does not choose to use these in their analyses, they should reference this literature and acknowledge that they do not use these variables.  

From the conclusion, it’s not completely clear what the impacts from this study are for the field of residential segregation. Make sure that readers are walking away knowing exactly what your contributions are. I would also suggest the author tie these findings back to the literature to strengthen their conclusions.

 I appreciated the maps and figures and found that they really added to the text and my understanding of the findings.

Round 2

Reviewer 2 Report

The author has done a good revision and solved most of my previous concerns related to the content of the paper. I think a minor revision is further required in terms of engaging with more literature in segregation and diversity studies. The weak literature engagement presented in this paper is reflected also in the limited sources from the reference list - there are only 23 cited references! Usually, academic papers have over 40-50 references. So I think the author can add several sentences mainly in the subsection 3.1. Residential segregation and present a little bit their content. I had some suggestions in my previous review, but there could be added many other studies too, so that to better position this study in the broader urban segregation and diversity literature. For instance, place attachment, segregation and marginalization is a nexus which can be shortly mentioned or even social mixing and (de)segregation is a hot topic in urban studies.  

Reviewer 3 Report

While I appreciate the author's clarification in the response memo, I would have liked to see them address these comments in the manuscript itself or edit their original submission so these questions won't arise from readers rather than essentially discount them and send the paper right back with minor edits. 

Round 3

Reviewer 3 Report

I appreciate the additional revisions that the author has made. These changes strengthen their paper and better situate it within the literature.